# Depletion of HuR in murine skeletal muscle enhances exercise endurance and prevents cancer-induced muscle atrophy

Brenda Janice Sánchez[1,2], Anne-Marie K. Tremblay[1,2], Jean-Philippe Leduc-Gaudet [3], Derek T. Hall [1,2], Erzsebet Kovacs[2], Jennifer F. Ma [1,2], Souad Mubaid[1,2], Patricia L. Hallauer[4], Brittany L. Phillips[5], Katherine E. Vest[6], Anita H. Corbett [5], Dimitris L. Kontoyiannis[7,8], Sabah N.A. Hussain[3], Kenneth E.M. Hastings[4], Sergio Di Marco[1,2] & Imed-Eddine Gallouzi[1,2]

The master posttranscriptional regulator HuR promotes muscle fiber formation in cultured muscle cells. However, its impact on muscle physiology and function in vivo is still unclear. Here, we show that muscle-specific HuR knockout (muHuR-KO) mice have high exercise endurance that is associated with enhanced oxygen consumption and carbon dioxide production. muHuR-KO mice exhibit a significant increase in the proportion of oxidative type I fibers in several skeletal muscles. HuR mediates these effects by collaborating with the mRNA decay factor KSRP to destabilize the *PGC-1α* mRNA. The type I fiber-enriched phenotype of muHuR-KO mice protects against cancer cachexia-induced muscle loss. Therefore, our study uncovers that under normal conditions HuR modulates muscle fiber type specification by promoting the formation of glycolytic type II fibers. We also provide a proof-of-principle that HuR expression can be targeted therapeutically in skeletal muscles to combat cancer-induced muscle wasting.

[1] Department of Biochemistry, McGill University, 3655 Promenade Sir William Osler, Montreal, QC H3G1Y6, Canada. [2] Rosalind & Morris Goodman Cancer Research Center, McGill University, 3655 Promenade Sir William Osler, Montreal, QC H3G1Y6, Canada. [3] Meakins-Christie Laboratories, Translational Research in Respiratory Diseases Program, and Department of Critical Care, McGill University Health Centre Research Institute, Montréal, QC, Canada. [4] Montreal Neurological Institute, McGill University, Montréal, QC, Canada. [5] Department of Biology, Emory University, Atlanta, GA, USA. [6] Department of Molecular Genetics, Biochemistry, & Microbiology, University of Cincinnati College of Medicine, 231 Albert Sabin Way, Cincinnati, OH 45267, USA. [7] Biomedical Sciences Research Centre "Alexander Fleming", Institute of Fundamental Biomedical Research, 16672 Vari, Greece. [8] Aristotle University of Thessaloniki, School of Biology, Department of Genetics, Development & Molecular Biology, 54124 Thessaloniki, Greece. Correspondence and requests for materials should be addressed to I.-E.G. (email: imed.gallouzi@mcgill.ca)

The importance of skeletal muscle is underscored by its requirement for locomotion, posture, and breathing and by the fact that loss of muscle function and integrity can lead to crippling and deadly consequences[1,2]. Many cancers trigger rapid muscle wasting, a condition also known as cachexia, that in turn leads to resistance to treatment, low quality of life and death[3].

Muscle fibers can be classified into two categories. Type I fibers are slow-contracting and are specialized for oxidative energy metabolism, having high levels of mitochondria and oxidative enzymes, and low levels of glycolytic enzymes than are found in type II fibers. Type II fibers are fast-contracting and are subdivided into three types. Type IIB are specialized for glycolytic metabolism, having high levels of glycolytic enzymes and low mitochondrial content. Type IIA are not metabolically specialized, having higher glycolytic enzyme levels than type I and higher mitochondrial content than type IIB. Type IIA and type I generate less force but are more resistant to fatigue in comparison with IIB fibers. Type IIX are intermediate between type IIA and type IIB in metabolic and contractile properties[1,4]. Each one of these fiber types expresses a unique isoform of the myosin heavy chain (MyHC) protein (Type I, IIA, IIX, and IIB respectively)[1,4]. Each individual muscle is composed of a mixture of various fiber types[4]. This heterogeneity in fiber type enables different muscle groups to achieve a variety of functions and movements[4]. Owing to their distinctive physiological and metabolic characteristics, fiber types are also differentially sensitive to specific pathophysiologic assaults. In pre-clinical mouse models of muscle-loss diseases, such as Duchenne Muscular Dystrophy (DMD) and cancer cachexia, Type II fibers are more prone to wasting when compared to Type I fibers[1,4,5]. Therefore, factors regulating fiber type in muscle could represent ideal drug targets for treating cachexia and other muscle wasting diseases.

It is well-established that factors such as the transcription factor peroxisome-proliferator-activated receptor gamma coactivator-1 alpha (PGC-1α) and the deacetylase Sirtuin 1 (Sirt1), modulate the fiber-type composition of skeletal muscles. Sirt1 enhances PGC-1α activity and together they promote the formation of oxidative (Type I) fibers[1,4]. In response to a stimulus such as voluntary exercise, the activation of Sirt1 leads to the deacetylation of PGC-1α. This in turn, upregulates the expression of NRFs (nuclear respiratory factors) and Tfam (mitochondria transcription factor A), which are key players in mitochondrial biogenesis and oxidative metabolism in muscles[1,4,6,7]. On the other hand, transcription factors such as Sineoculis homeobox homolog 1 (Six1) and nuclear factor of activated T-cells, cytoplasmic4 (NFATc4) modulate the expression of target genes involved in the formation of glycolytic type II or IIX fibers[1,4]. In addition to the specific genes activated by these factors, their impact on fiber-type specification also depends on their level of expression in response to various stimuli[1,4]. Therefore, the molecular mechanisms controlling the expression levels of these factors play a crucial role in muscle fiber-type specification under different conditions.

Although the expression of these and other factors is modulated transcriptionally[1,4], some observations have established that post-transcriptional regulatory mechanisms also affect their levels in response to exercise. A decrease in the half-life of *PGC-1α* and *TFAM* messenger RNAs (mRNAs) by ~40–60% in slow-twitch oxidative muscles correlates with an increase in the expression levels of RNA-binding proteins (RBPs) such as HuR and the mRNA decay factor KSRP[8]. While a role for HuR and KSRP in the regulation of these and other mRNAs during exercise is still elusive, the involvement of HuR and KSRP in the formation of muscle fibers in cell culture is well-established[9–13]. During the early stages of myogenesis, HuR both promotes the translation of the *HMGB1* mRNA[11] and collaborates with KSRP to reduce the expression of nucleophosmin (NPM) protein by destabilizing the *NPM* mRNA[12]. At later steps of myogenesis, however, HuR stabilizes the mRNAs encoding promoters of muscle fiber formation such as *MyoD*, *Myogenin*, and *p21*, only when muscle cells begin their fusion to form fibers (myotubes)[10].

To investigate the in vivo relevance of HuR in muscle tissues, in this study we use the Cre-LoxP system to generate a HuR muscle-specific knockout mouse (*MyoDCre⁺;Elavl1ᶠˡ/ᶠˡ*). We show that the loss of HuR leads to the enrichment of type I fibers resulting in the increased oxidative metabolic capacity of the skeletal muscle. This indicates that one of the main roles of HuR in skeletal muscles is to promote the formation and maintenance of glycolytic type II fibers. HuR mediates these effects by destabilizing the *PGC-1α* mRNA in a KSRP-dependent manner. We also provide data demonstrating that depleting the expression of HuR in muscles protects mice against cancer-induced muscle atrophy.

## Results

**HuR depletion in muscle improves endurance and oxidative capacity.** The total knockout of the HuR gene (also known as *Elavl1*) is embryonic lethal (embryos die between E10.5-E14.5)[14]. We therefore generated an *Elavl1* muscle-specific knockout (muHuR-KO) mouse to investigate the in vivo role of HuR in muscle formation and muscle physiology. Mice carrying the *Elavl1ᶠˡ/ᶠˡ* allele[14] and mice expressing Cre recombinase under the control of the *MyoD* promoter[15] were bred to obtain the HuR muscle-specific knockout (Fig. 1a). The knockout of HuR is initiated in muscle progenitor cells during embryogenesis, since Cre under the *MyoD* promoter is activated in the branchial arches and limb buds as early as day E10.5[15].

muHuR-KO mice are viable and do not exhibit any major change in their total body weight (Fig. 1b, c). Knockout of HuR was confirmed by genotyping with PCR primers and by western blot (WB) analysis in several hindlimb skeletal muscles, including the *quadricep*, *gastrocnemius*, *tibialis anterior* (TA), *soleus*, *peroneus*, and *extensor digitorum longus* (EDL) (Fig. 1d–f). The fact that muHuR-KO mice are healthy with no obvious defect suggests that, in vivo, the role of HuR in the formation, development and function of skeletal muscles is either redundant with other RBPs (see discussion) or that HuR-mediated regulation is more relevant in post-natal muscle development during adaptation to various muscle-related functions and needs.

To investigate the above-mentioned possibilities, we assessed muscle-related functions in muHuR-KO compared to control mice. To do this, we used invasive and non-invasive in vivo tests: in situ analysis of muscle contractility, which measures force generation and fatigability[16,17], the treadmill exhaustion test, which estimates exercise capacity and endurance, and the limb grip strength assays, which determines muscle strength[18]. In situ analysis showed that although TAs of muHuR-KO mice exhibited a higher contraction force than those of control animals, they did not demonstrate any notable differences in the fatigability test (Fig. 2a). Additionally, a treadmill exhaustion test indicated that the time to exhaustion and the running distance covered by muHuR-KO mice was significantly longer than their control counterparts (Fig. 2b, c). In this test, muHuR-KO mice performed 20% more work than the control mice (Fig. 2d). Of note, this increase in endurance was accompanied by a slight decrease in muscle strength of the muHuR-KO mice (Fig. 2e and Supplementary Fig. 1). We also confirmed increased exercise endurance in the muHuR-KO mice using the accelerating Rota-rod and the Inverted-grid platform (Fig. 2f, g).

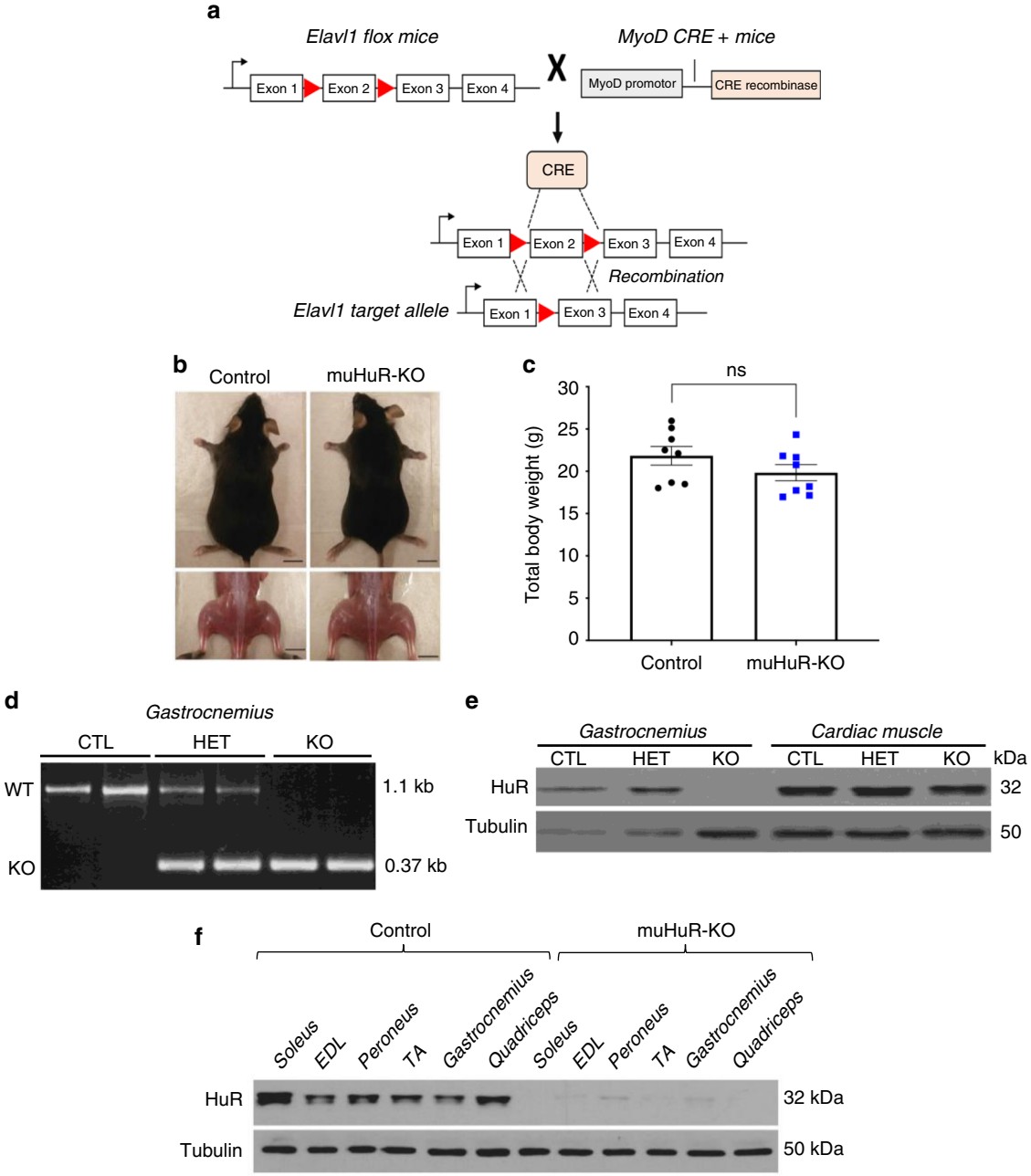

**Fig. 1** Generation of HuR muscle-specific knockout mice using the Cre-lox P system. **a** Diagram depicting the tissue-specific knockout strategy. Elavl1-flox mice (Lox P sites ►) were breed with mice expressing Cre recombinase under the control of the MyoD promoter (MyoD CRE[+]) to generate muscle-specific HuR KO mice. **b** Photographs of 2-month-old muHuR-KO and control male mice. Scale bars = 1 cm. **c** Total body weights of 8–10-weeks-old muHuR-KO and control mice ($n = 8$). The results are presented as mean ± S.E.M, *$p < 0.05$ unpaired $t$-test. **d** PCR amplification of the targeted region of the *Elavl1* gene in *gastrocnemius* muscle samples from control (CTL), heterozygote (HET), and muHuR-KO (KO) mice. Shown is a representative of agarose gel of the genotyping of all the mice used in this study ($n = 30$). **e** Representative western blot analysis, from four independent experiments, of HuR expression in skeletal and cardiac muscle tissue from control CTL, HET, and KO mice using antibodies against HuR or α-tubulin. **f** Representative western blot analysis of *soleus, extensor digitorum longus* (EDL), *peroneus, tibialis anterior* (TA), *gastrocnemius,* and *quadriceps* muscles from control and muHuR-KO mice using antibodies against HuR or α-tubulin. This blot is a representative of four independent experiments. Source Data are provided in the Source Data File

Enhanced endurance is generally associated with an increased oxidative capacity of skeletal muscle fibers[1,4,19]. Therefore, we used the Columbus Instrument's Comprehensive Animal Monitoring System (CLAMS)® (Fig. 3a) to determine the rate of oxygen consumption ($VO_2$) and carbon dioxide production ($VCO_2$), two indicators widely used to measure the oxidative capacity of rodents and humans[20]. While we did not observe any change in the voluntary movement of these animals (Supplementary Fig. 2a), muHuR-KO mice showed a higher rate of $VO_2$ consumption and $VCO_2$ production than their control littermates (Fig. 3b, c). The muHuR-KO animals exhibited a slight increase in the respiratory exchange ratio (RER), that is most evident during the peak of voluntary movement (Fig. 3d), suggesting that, at least under non-exercise conditions, the depletion of HuR favors the usage of carbohydrate as a source of energy in skeletal muscles[21]. On the other hand, several key

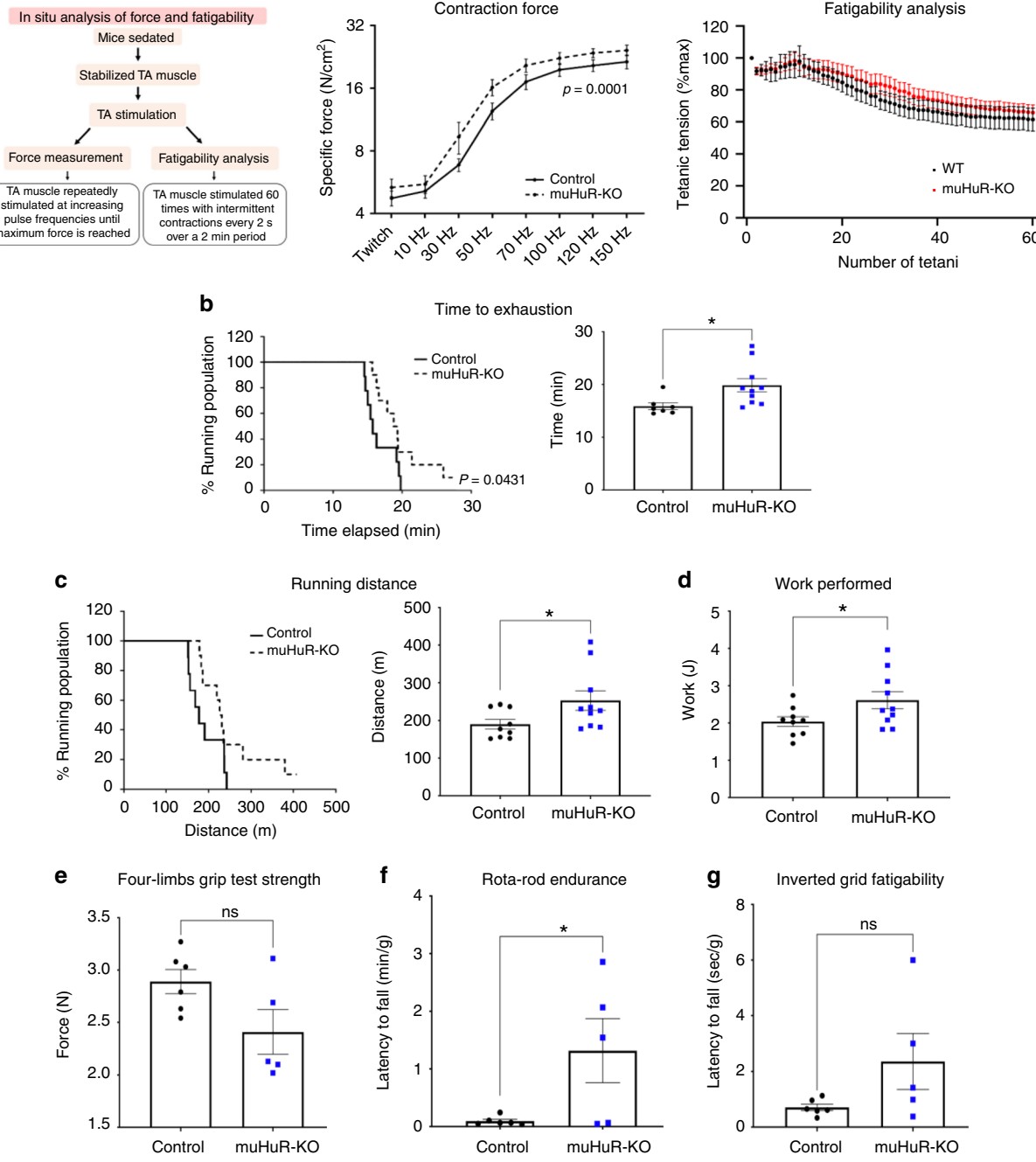

**Fig. 2** HuR muscle-specific KO mice have enhanced exercise endurance. **a** Left panel: schematic describing the method used to determine, in situ, force measurement, and fatigability of control and muHuR-KO mice. Middle panel: contractile function of the TA muscle was assessed in situ at various stimulation frequencies (Control: $n = 8$, muHuR-KO: $n = 6$). Right panel: fatigability of TA muscle was assessed in situ over 60 stimulation sessions with a resting period of 2 min between stimuli. Fatigability was normalized to TA muscle weight shown in (Supplementary Fig. 1a) (Control: $n = 8$, muHuR-KO: $n = 6$). **b–g** Physical performance was evaluated in age-matched control and muHuR-KO mice by performing a treadmill exhaustion test. Three parameters were measured with this test: **b** Time to exhaustion (left panel; survival plot showing the percentage of mice running at indicated time points. Right panel; mean duration of the run). **c** Running distance (left panel; survival plot showing the percentage of mice running at indicated distances. Right panel; mean distance ran) and **d** Work performed during test. [**b** Control: $n = 7$, muHuR-KO: $n = 10$, **c** control: $n = 9$, muHuR-KO: $n = 10$, **d** control: $n = 9$, muHuR-KO: $n = 10$]. **e** Grip strength was evaluated on control and muHuR-KO mice using a digital force gauge. Mice were allowed to grip using their four limbs (forelimb and hindlimbs) and peak force was measured in triplicate during two sessions (Control: $n = 6$, muHuR-KO: $n = 5$). **f** Endurance performance was assessed using a Rota-rod system. For each animal, the duration on the rod was measured twice with a resting period of 4 days in between. The latency to fall represents an average of the two sessions of evaluation (Control: $n = 6$, muHuR-KO: $n = 5$). **g** Fatigability was evaluated by performing an inverted-grid test, the latency to fall represents an average of two sessions of evaluation normalized to total body weight (Control: $n = 6$, muHuR-KO: $n = 5$). Statistical analysis for in situ data shown in **a** was performed using two-way ANOVA. The results are presented as mean ± S.E.M, *$p < 0.05$ unpaired $t$-test **b–g** Source Data are provided in the Source Data File

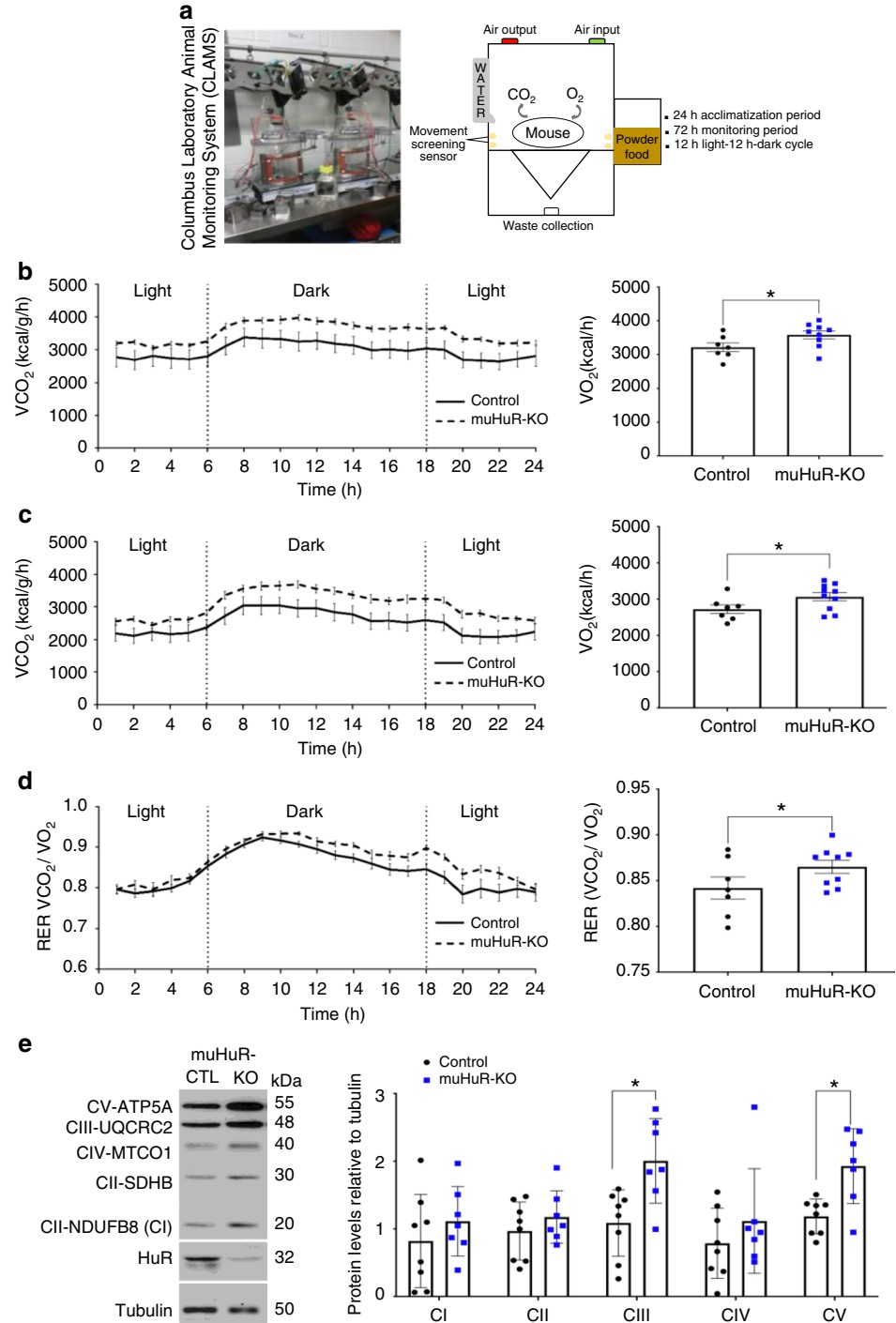

**Fig. 3** muHuR-KO mice show an increased oxidative capacity. **a** Schematic illustrating the Comprehensive Animal Monitoring System (CLAMS; Columbus Instruments, Columbus, OH) that was used to complete a 3-day indirect calorimetry study in age-matched mice under a 12 h light–12 h dark cycle. **b**, **c** Oxygen consumption ($\dot{V}O_2$) (**b**), and carbon dioxide production ($\dot{V}CO_2$) (**c**) were measured over the 3 days following a 24-h acclimation period. **d** Respiratory exchange ratio (RER) was calculated as the ration of $\dot{V}CO_2/\dot{V}O_2$. The graphs on the left depicts the average values at each time point while that on the right shows the average or cumulative (only for activity) values over the 72 h period. **b**–**d** Data obtained was analyzed using the CLAMS examination tool (CLAX; Columbus Instruments) version 2.1.0. (Control: $n = 7$, muHuR-KO: $n = 9$). The results are presented as mean ± S.E.M, *$p < 0.05$ unpaired $t$-test. **e** Left panel: western blot analysis of levels of OXPHOS complexes in the mitochondria of control and muHuR-KO mice. Right panel: quantifications of the levels of the complexes are presented as the mean ± S.E.M, *$p < 0.05$ unpaired $t$-test (control: $n = 8$, muHuR-KO: $n = 7$). Source Data are provided in the Source Data File

components of the electron transport chain (ETC) complexes such as CIII and CIV, as well as the ATP synthase CV, a well-established indicators of mitochondrial oxidative respiration[22], show a significant increase in their expression level in muHuR-KO muscles (Fig. 3e). The fact that the absence of HuR did not have any effect on heat production levels (Supplementary Fig. 2b), suggested that the oxidative phenotype observed in muHuR-KO mice is not associated with metabolic uncoupling[23]. Therefore, overall our results indicate that the specific disruption of the *HuR* gene in muscle improves exercise endurance and oxygen consumption.

**Loss of HuR in muscle promotes type I oxidative fibers.** Improved endurance is, in general, associated with a noticeable increase in the proportion of type I fibers[24]. Hence, we examined the fiber-type composition of several muscles isolated from control and muHuR-KO mice by performing metachromatic ATPase staining with specific antibodies against Myosin Heavy Chain (MyHC) I, IIA, and IIB. The *soleus* of muHuR-KO mice showed a ~17% enrichment of Type I fibers when compared to control animals, while Type IIA fibers decreased by ~16% (Fig. 4a–c and Supplementary Table 1). In keeping with this, the *soleus* of muHuR-KO mice showed an increase in the steady-state levels of mRNAs encoding some promoters of type I-fibers such as *Tnnl1* and *MyHC I*, and a decrease in promoters of type II-fibers such as *Tnnt3* and *MyHC IIB* (Fig. 4d). The same effects on the proportion of fiber Type I and on the expression modulators of fiber-type specification were also observed in both the *peroneus* and EDL of the muHuR-KO mice (Supplementary Figs. 3 and 4 and Supplementary Table 1). We also carried out a muscle fiber size analysis by measuring cross-sectional area and found that the distribution pattern of fiber size in the *soleus* was not affected by HuR loss (Fig. 4e–g). Thus, the observed change in fiber-type composition is not associated with a defect in muscle fiber generation or growth. Altogether these findings demonstrate the involvement of HuR in the regulation of muscle fiber-type composition in vivo, where the presence of HuR favors the formation of type II, while its depletion promotes the formation of type I.

**HuR depletion in muscle activates PGC-1α and its associated pathway.** To delineate the molecular mechanisms through which HuR modulates fiber-type specification in mice, we performed a high-throughput mRNA sequencing (RNA-seq) analysis. Total RNA was isolated from *soleus* muscles of muHuR-KO and control littermates and was used to prepare and sequence four mRNA libraries[25]. The clear separation of the two genotypes was evident in the heatmap showing the 17,534 genes detected through RNA-seq analysis in both control and muHuR-KO mice (Supplementary Fig. 5a). RNAseq-data was further examined using a DESeq2 package for differential expression analysis. A volcano plot of the acquired data shows the general profile of gene expression and highlights the genes that are up (right side) or downregulated (left side) in muHuR-KO muscles when compared to control counterparts (Fig. 5a). Each dot represents a single gene while the horizontal and vertical dashed lines indicate the statistically significance threshold (log2FC > 0.5 or < −0.5, *p* = 0.05). From the 1914 genes affected in the *soleus* muscle of HuR knockout mice (1.5-fold change or more), 86% were increased, while only 14% were decreased (Supplementary Data 1). Of note, the steady level of well-known HuR mRNA targets in muscle fibers, such as *MyoD*, *Nucleophosmin (NPM)* and *HMGB1*, were as expected, decreased (*MyoD*), increased (*NPM*) or remained unaffected (*HMGB1*) (Fig. 5b)[9–12]. The fact that the majority of the affected transcripts in the absence of HuR are increased, raises

the possibility that, in vivo, HuR has an overall destabilizing activity on its mRNA targets in skeletal muscle.

Next, to identify the pathways affected by the loss of HuR, we performed a core analysis using the Ingenuity Pathway Analysis software (IPA: Ingenuity Systems®). Two of the top five canonical pathways identified by IPA were associated with the activity of the transcription factor, *Peroxisome proliferator-activated receptor alpha* (PPARα) (Fig. 5c and Supplementary Fig. 5b). PPARα plays a critical role in energy production and lipid and carbohydrate metabolism and also regulates the expression of genes involved in peroxisomal and mitochondrial β-oxidation[4]. A reduction in the expression levels or a complete deletion of the PPARα gene are associated with an increased endurance capacity of animals. On the other hand, PPARβ/δ collaborate with PGC-1α to promote oxidative phenotype and increase endurance capacity of skeletal muscles[4].

To validate our IPA analysis, total RNA from *soleus*, *peroneus* and EDL muscles of both control and muHuR-KO mice was prepared and used to determine the transcript levels of genes involved in the PPARα signaling pathway or fiber-type specification such as *PGC-1α, PGC-1β, Tfam, PPARα, Six1* (Sineoculis homeobox homolog 1), *NCOA6* (Nuclear receptor coactivator 6), *Tpm1* (Tropomysin 1), and *MyoD*[1,4]. Consistent with the RNAseq data and the observed type I fiber enrichment phenotype, we observed a twofold increase in the steady-state level of both *PGC-1α* mRNA and protein in the *soleus* of muHuR-KO mice when compared to control littermates (Fig. 5d, e). However, although maintained, the increase in *PGC1α* expression level was less drastic in the *peroneus* and EDL of muHuR-KO mice (Supplementary Fig. 6). HuR loss, on the other hand, had a very little effect on the steady-state level of *PGC-1β* and *Tfam* mRNAs, two factors associated with the oxidative phenotype (Fig. 5d and Supplementary Fig. 6). In addition, loss of HuR not only decreased, as expected[9,12], the steady-state level of *MyoD* mRNA, but also the levels of mRNAs encoding other factors involved in the glycolytic phenotype such as *PPARα, Six1* and *Tpm1* (Fig. 5d and Supplementary Fig. 6)[4].

It is well-established that the induction of an oxidative phenotype in muscles is, in general, associated with an increase in fatty acid oxidation and mitochondria biogenesis and function[1,4]. While muHuR-KO muscle did not exhibit any change in the expression levels of genes involved in fatty acid transport and oxidation, such as *Acadv1, CD36, FAS, LDL, UCP2,* and *UCP3*[1,4], it did show an increase in the level of *NRF1* (nuclear respiratory factor 1) mRNA (Supplementary Fig. 7b), a known promoter of mitochondrial oxidative respiration in various tissues including muscle[4]. However, the ratio between mitochondrial (*mtDNA*) and nuclear (*nDNA*) DNA was unchanged in both muHuR-KO muscles and their control counterparts (Supplementary Fig. 7c), indicating that while the depletion of *HuR* gene in muscles does not affect mitochondrial biogenesis, it could be associated with an enhancement of mitochondrial activity.

Altogether, these results show that by regulating the expression levels of key genes such as *PGC-1α*, HuR controls energy metabolism and fiber-type specification of skeletal muscles.

**HuR collaborates with KSRP to destabilize PGC-1α mRNA in muscle cells.** PGC-1α is one of the major promoters of type I oxidative phenotype in skeletal muscle[4]. As a first step in determining the way by which HuR regulates PGC-1α expression, we investigated whether HuR binds to the *PGC-1α* mRNA in muscle cells. Consistent with previous observations[26], we were unsuccessful in immunoprecipitating HuR from skeletal muscle extracts. Therefore, we used the well-established C2C12 myoblasts, to assess, as we did before[9,11,12,27], the association between

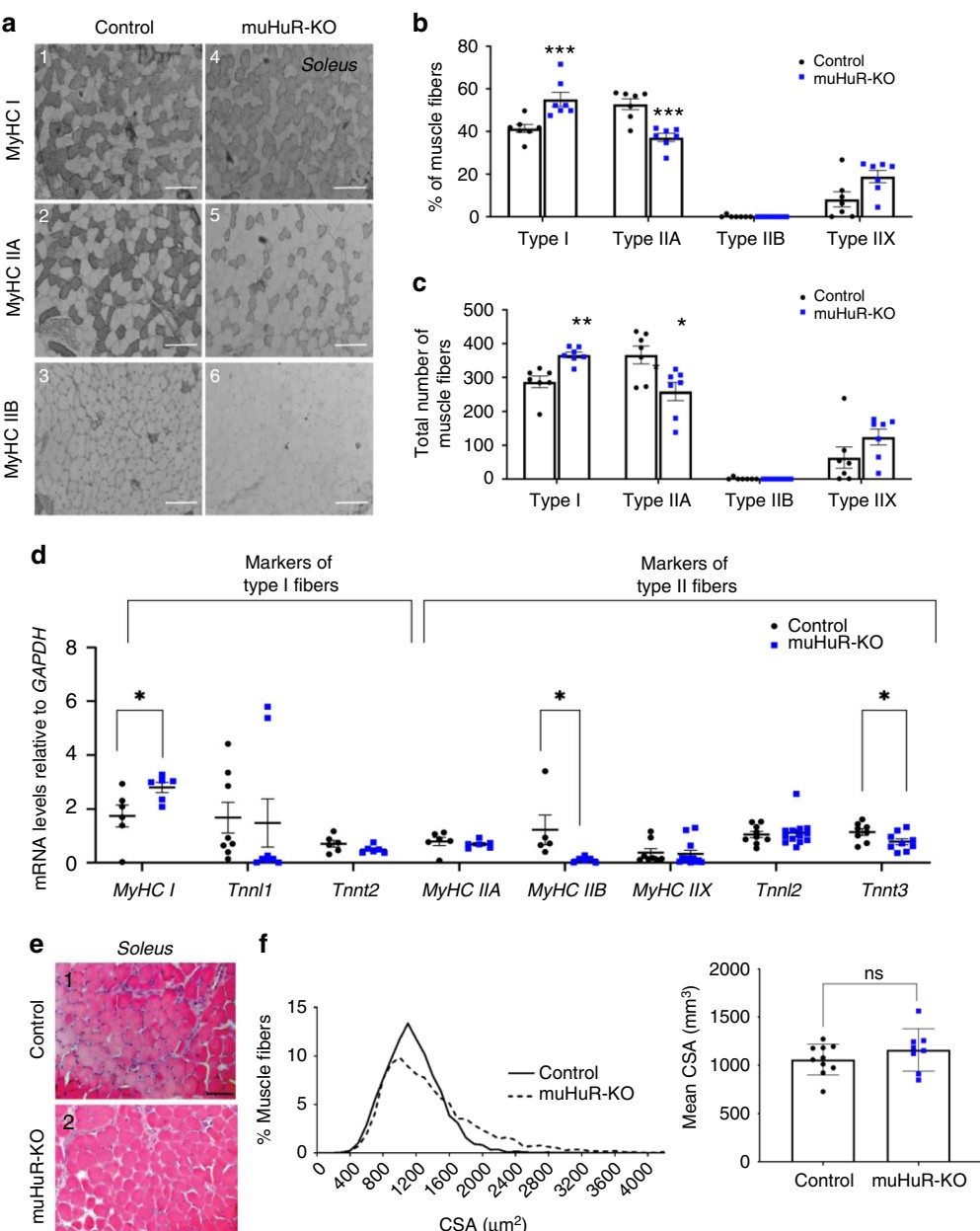

**Fig. 4** Depletion of HuR in skeletal muscle increases the proportion of type I fibers. **a** Representative photomicrographs of serial sections of *soleus* muscles from control and muHuR-KO mice taken after immunostaining with anti-Myosin Heavy Chain (MyHC) antibodies type I, type IIA, and type IIB. Scale bars: 100 μm. Photomicrographs are representative of sections prepared from seven different mice. **b** Quantification of muscle fibers type I, type IIA, and type IIB was ascertained manually. Fibers type IIX were calculated by counting the unstained fibers. Results are graphed as a percentage of the total number of fibers per muscle. **c** Total number of fibers per muscle is shown for the muscles analyzed in **b**. *n* = 7 mice for **b**, **c**. **d** mRNA expression of known markers of fiber-type specificity, *MyHC I*, *Tnnt2*, and *MyHC IIA* (*n* = 6 mice), *Tnnl1* (*n* = 8 mice), *MyHC IIB* (*Control n* = 5, *muHuR-KO n* = 6 mice), *MyHC IIX* (*Control n* = 8, *muHuR-KO n* = 12 mice), *Tnnl2* (*Control n* = 9, *muHuR-KO n* = 13 mice), and *Tnnt3* (*Control n* = 8, *muHuR-KO n* = 9 mice) was assessed by RT-qPCR. mRNA levels were standardized against *GAPDH* and plotted relative to the expression in control mice. **e** Representative photomicrographs of *soleus* muscles sections from control and muHuR-KO mice taken after H&E staining. Scale bars: 100 μm (*n* = 8 mice). **f** Left panel: mean CSA of *soleus* muscles fibers from control and muHuR-KO mice were analyzed from sections stained with H&E. Right panel: frequency histogram showing the distribution of muscle fiber CSA in the *soleus* muscles from control and muHuR-KO mice. (control: *n* = 10, muHuR-KO: *n* = 8). Results are presented as mean ± S.E.M, *$p < 0.05$, **$p < 0.01$, ***$p < 0.001$ unpaired *t*-test. Source Data are provided in the Source Data File

HuR and *PGC-1α* mRNA. Similar to what was observed in the *soleus*, small-interfering ribonucleic acid (siRNA)-mediated HuR depletion in C2C12 myoblasts[11,12] significantly increased *PGC-1α* mRNA and protein levels (Fig. 6a, b). Immunoprecipitation of HuR from these myoblasts coupled with reverse transcription quantitative PCR (RT-qPCR) analysis revealed that HuR forms a complex with the *PGC-1α* mRNA (Fig. 6c).

Next, we determined the post-transcriptional level through which HuR regulates *PGC-1α* mRNA expression. To test mRNA translation, we performed polysome fractionation experiments on C2C12 myoblasts depleted or not of HuR, using siRNA control (siCtl) or against HuR (siHuR), and followed the recruitment of *PGC-1α* mRNA to heavy polysomes, a well-established assay used to identify actively translated messages[12]. We observed no

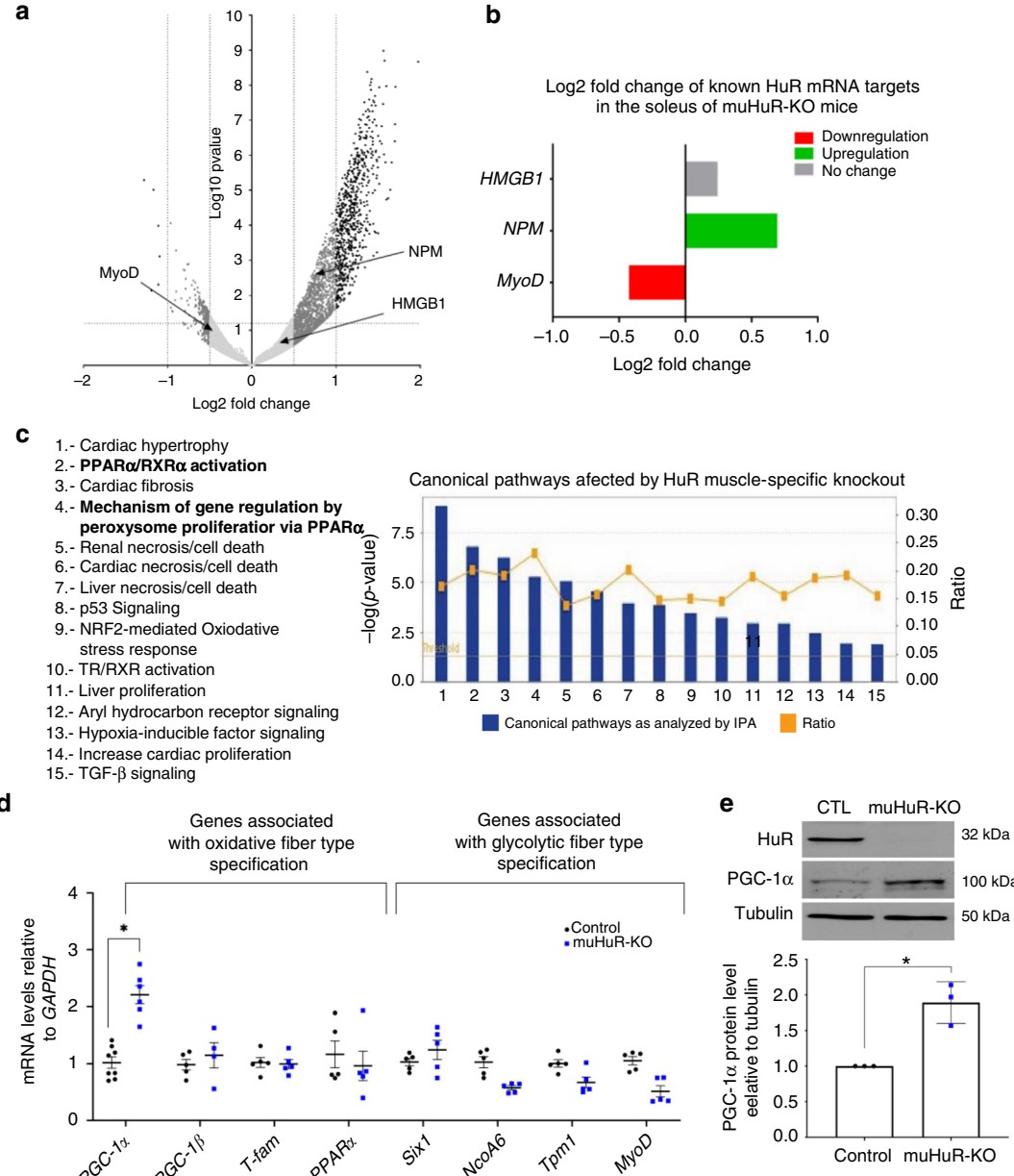

**Fig. 5** Increased PGC-1α expression in muHuR-KO muscle. **a** Volcano plot showing the log2 fold-difference in mRNA expression in the *soleus* muscle of control and muHuR-KO mice as assessed by the DESeq2 analysis of the RNA-seq data. Negative values indicate decrease in gene expression while positive values refer to upregulation of gene expression. Dash lanes indicate threshold for statistical significance ($p = 0.05$ for log2FC > 0.5, < 0.5). The location of known HuR mRNA targets, including *NPM*, *HMGB1*, and *MyoD* are shown. **b** Comparison of log2 fold change score of the previously identified HuR mRNA targets *NPM*, *HMGB1*, and *MyoD* is shown. **c** Bar graph indicating the signaling pathways affected by the knockout of HuR in *soleus* muscle as analyzed by Ingenuity Pathway Analysis software (IPA®). The x-axis represents the identified pathways. The y-axis (left) shows the −log10 of the p-value. The ratio (y-axis, right) represented by the orange points is calculated as follows: numbers of genes in each pathway that meet cutoff criteria, divided by total numbers of genes that are involved in that pathway. The horizontal orange line indicates the threshold above which there is statistical significance. **d** Total RNA was isolated from *soleus* muscles of control and muHuR-KO mice and relative expression level of genes associated with PPAR signaling and/or fiber-type specification (*PGC-1α, PGC-1β, Tfam, PPARα, Six1, NCOA6, Tpm1, MyoD*) was assessed by RT-qPCR. Relative mRNA levels were standardized against *GAPDH* and plotted relatively to the expression in control mice ($n = 5$ mice all the genes except for *PGC-1α*, were control $n = 8$ and muHuR-KO $n = 6$). **e** Western blot (top panel) and relative quantification (bottom panel) of PGC-1α protein levels in *soleus* muscle from control and muHuR-KO mice using antibodies against PGC-1α, HuR, or α-tubulin ($n = 3$). The results are presented as mean ± S.E.M, *$p < 0.05$ unpaired t-test. Source Data for panels **d**, **e** are provided in the Source Data File and raw data for RNAseq are provided in Supplementary Data 1

difference in the levels of *PGC-1α* mRNA in heavy polysomes in the presence or absence of HuR (Supplementary Fig. 8). Actinomycin D pulse-chase experiments[12,28] were used to determine if HuR regulates the stability of the *PGC-1α* mRNA in these cells. Knocking down HuR in myoblasts increased the

half-life of *PGC-1α* mRNA from 6 to > 10 h (Fig. 6d). As expected[12], however, loss of HuR destabilized known mRNA targets of HuR such as *MyoD* (Fig. 6e) and *Myogenin* (Supplementary Fig. 9). On the other hand, C2C12 myoblasts overexpressing GFP-HuR exhibited a significant decrease in the

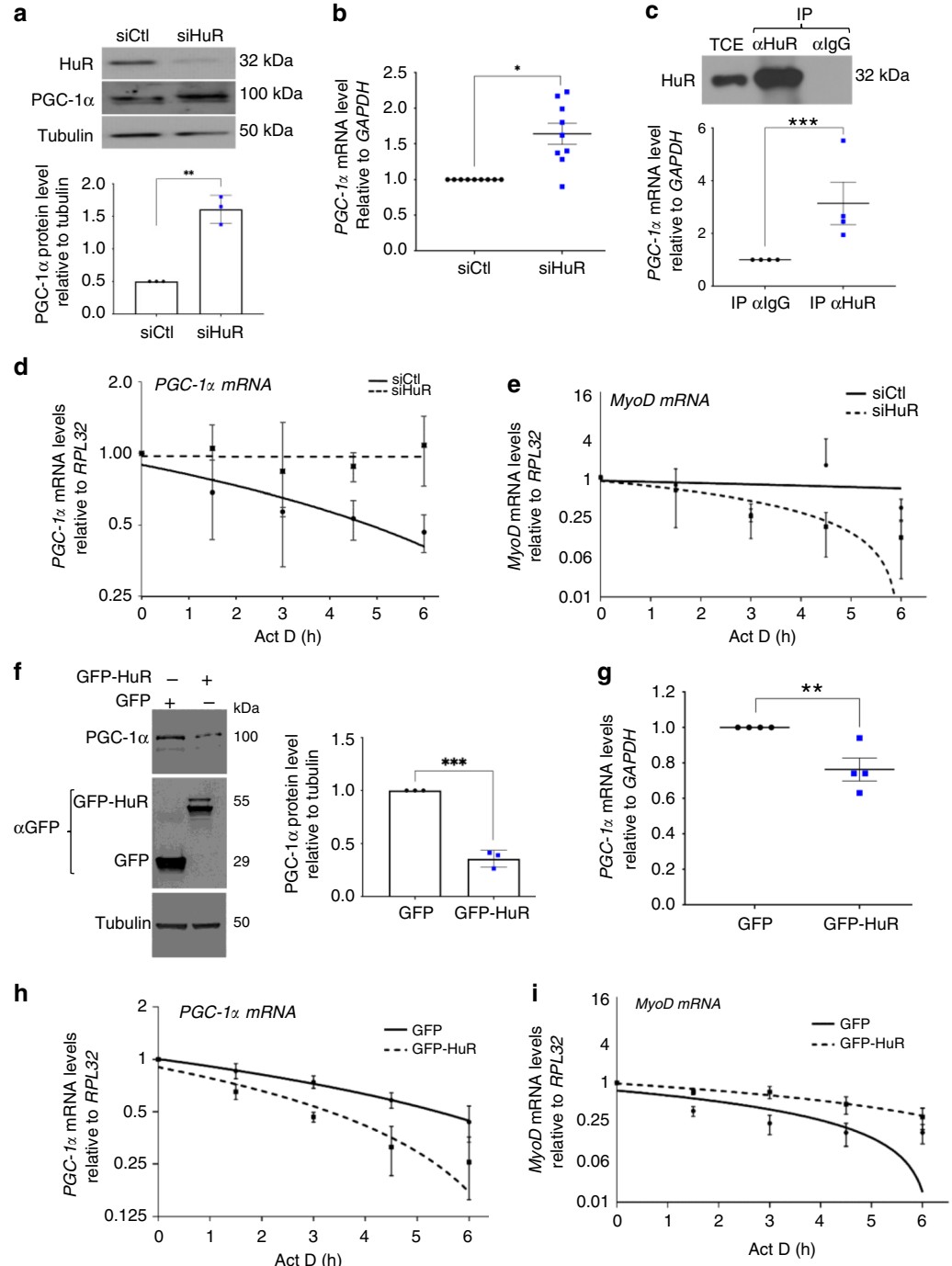

**Fig. 6** HuR destabilizes the PGC-1α mRNA in muscle cells. **a** Western blot (top panel) and relative quantification (bottom panel) of PGC-1α protein levels in myoblasts treated with or without siHuR using antibodies against PGC-1α, HuR, or α-tubulin (n = 3). **b** Total RNA was isolated from myoblasts treated as in **a** and *PGC-1α* expression was assessed by RT-qPCR. *PGC-1α* mRNA level were standardized against *GAPDH* and plotted relatively to siCtl (n = 9). **c** Top panel: western blot showing the immunoprecipitation (IP) of HuR using an anti-HuR antibody (3A2) or IgG as control. (Bottom panel) RT-qPCR were used to determine the association of the *PGC-1α* mRNA to HuR. Normalized *PGC-1α* mRNA levels were plotted relatively to the IgG control (n = 4). **d**, **e** The stability of the *PGC-1α* (**d**) and *MyoD* mRNAs (**e**) was determined in myoblasts depleted (siHuR) or not (siCtl) of HuR and treated with Actinomycin D (ActD) for 0, 1.5, 3, 4.5, or 6 h (**h**). mRNA levels were then standardized against *RPL32* mRNA levels and plotted relative to the abundance of mRNA at time 0 of ActD treatment (which is represented as 1) (n = 3). The line of best fit was determined by linear regression using the data points for siCtl and siHuR. Error bars represent ± S.E.M. **f** Western blot (left panel) and relative quantification (right panel) of PGC-1α protein levels in myoblasts expressing GFP or GFP-HuR using antibodies against PGC-1α, GFP or α-tubulin (n = 3). **g** Total RNA was isolated from myoblasts overexpressing or not HuR (GFP-HuR) and relative *PGC-1α* expression was assessed by RT-qPCR. *PGC-1α* mRNA level were standardized against *GAPDH* and plotted relatively to siCtl (n = 4). **h**, **i** Myoblasts expressing GFP or GFP-HuR were used to assess the stability of the *PGC-1α* (**h**) and *MyoD* (**i**) was determined as described in panels **d**, **e** (n = 3). For **d**, **e**, **h**, **i** the line of best fit was determined by linear regression using the data points for siCtl and siHuR. Error bars represent ± S.E.M. The results in **b**, **c**, **f**, **g** are presented as mean ± S.E.M, *p < 0.05, **p < 0.05, ***p < 0.005 unpaired *t*-test. Source Data for all panels are provided in the Source Data File

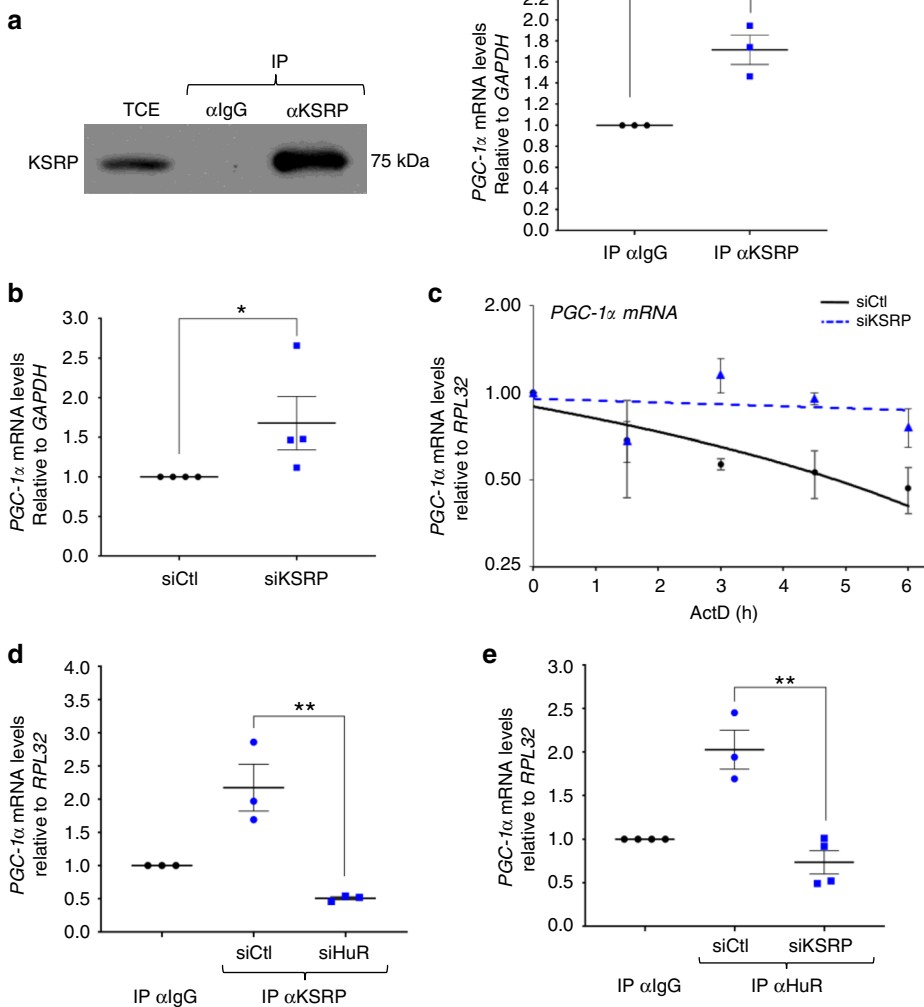

**Fig. 7** KSRP collaborates with HuR to destabilize the PGC-1α mRNA in muscle cells. **a** Left panel: western blot demonstrating the immunoprecipitation (IP) of KSRP using an anti-KSRP antibody or IgG as a negative control. Right panel: RT-qPCR experiments were performed to determine the association of *PGC-1α* mRNA to immunoprecipitated KSRP. Normalized *PGC-1α* mRNA levels were plotted relatively to the IgG negative control ($n = 3$). **b** Total RNA was isolated from C2C12 myoblasts treated with or without siKSRP and relative *PGC-1α* expression was assessed by RT-qPCR. *PGC-1α* mRNA level were standardized against *GAPDH* and plotted relatively to the siCtl condition ($n = 4$). **c** The stability of the *PGC-1α* mRNA was determined in muscle cells depleted or not of KSRP and treated with Actinomycin D (ActD) for 0, 1.5, 3, 4.5, or 6 h. mRNA levels were then standardized against *RPL32* mRNA levels and plotted relative to the abundance of mRNA at time 0 of ActD treatment (which is represented as 1) ($n = 3$). The line of best fit was determined by linear regression using the data points for siCtl and siKSRP. Error bars represent ± S.E.M. **d**, **e** IP coupled to RT-qPCR experiments was performed using anti-KSRP (**d**) or anti-HuR (**e**) antibodies on total extract from C2C12 myoblasts treated with siHuR (**d**), or siKSRP (**e**). *PGC-1α* mRNA levels in the samples immunoprecipitated with anti-KSRP or anti-HuR were normalized to the corresponding IgG sample. The results are presented as mean ± S.E.M, **$p < 0.01$ unpaired *t*-test. **d** ($n = 3$), **e** ($n = 4$ for IPIgGsiCtl, IPIgG siKSRP, and IPHuR siKSRP; $n = 3$ for IPHuR siCtl). Source Data for all panels are provided in the Source Data File

expression levels of PGC-1α protein and mRNA, as well as a ~40% reduction in the half-life of *PGC-1α* transcript (Fig. 6f–i). We also observed that the depletion of HuR does not affect the stability of other mRNAs involved in fiber-type specification such as *Tnnl1, Tnnl2, Six1, NFATc1,* and *NCOA6* (Supplementary Fig. 9). Collectively, these results strongly suggest that HuR antagonizes the type I fiber phenotype in muscle cells by destabilizing *PGC-1α* mRNA, leading to a decreased expression of PGC-1α protein.

Next, we examined the possibility that KSRP could be involved in the HuR-mediated destabilization of the *PGC-1α* mRNA. Immunoprecipitation experiment coupled with RT-qPCR analysis showed that KSRP, similarly to HuR, associates with the *PGC-1α* mRNA in myoblasts (Fig. 7a). Furthermore, knockdown of KSRP in these cells[12] significantly increased both the steady-state

level and the half-life of *PGC-1α* mRNA (Fig. 7b, c). We previously demonstrated that HuR and KSRP form a tight complex in C2C12 myoblasts and that the binding of HuR or KSRP to the *NPM* mRNA requires an intact HuR/KSRP complex[12]. Using similar experimental approaches, we observed that this is also the case for the binding of *PGC-1α* mRNA to either one of these RBPs (Fig. 7d, e). Therefore, one way by which HuR promotes the glycolytic phenotype in muscle cells and tissues is by destabilizing the *PGC-1α* mRNA in a KSRP-dependent manner.

**muHuR-KO mice are resistant to cancer-induced muscle wasting.** Several reports have suggested that at late stages, numerous cancers preferentially target type II glycolytic fibers to

trigger rapid muscle loss, a deadly condition also known as cachexia-induced muscle wasting[29]. Furthermore, the upregulation of PGC-1α expression has been associated with not only the promotion of type I oxidative fibers but also with the prevention of muscle atrophy induced by several conditions[30,31]. Therefore, we tested the possibility that the muHuR-KO mice could be protected from disease-induced muscle wasting.

To achieve this, we injected in control and muHuR-KO mice the Lewis Lung Carcinoma (LLC) cells, a cancer-cell model that is widely used to trigger muscle wasting in C57BL/6 mice[32,33]. Although, both control and muHuR-KO mice-bearing LLC tumors (LLC-Control and LLC-muHuR-KO) show no change in total body weight during the four weeks of tumor growth (Supplementary 10a), upon sacrifice the carcass weight minus tumor weight of the muHuR-KO mice demonstrated a significant protection from LLC-induced weight loss when compared to their control counterparts (Fig. 8a). Of note, control and muHuR-KO mice-bearing LLC tumors show no difference in tumor growth or tumor burden and exhibit comparable levels of systemic inflammatory response, as evidenced by the enlargement of the spleen, as well as by the loss of hindlimb fat pad (Supplementary Fig. 10b–e). Importantly, the atrophy of several hindlimb muscles, including the *gastrocnemius*, TA, *soleus* and *peroneus*, is significantly lower in LLC-muHuR-KO mice when compared to LLC-control mice (Fig. 8b). In addition, the expression levels of *atrogin-1/MAFbx* and *MuRF-1* mRNAs, two muscle-specific ubiquitin ligases that play an essential role in promoting cancer-induced muscle loss[34], is strongly induced (6-fold) in the muscle of LLC-control but not in LLC-muHuR-KO mice (Fig. 8c). Moreover, the high expression levels of *PGC1α* observed in muHuR-KO muscle was also maintained in the presence of LLC tumors (Fig. 8d). The expression levels of *MyHC I* was significantly reduced in control mice-bearing LLC tumors, while the high expression levels of MyHC I observed in the absence of HuR was maintained in muscles from LLC-muHuR-KO mice (Fig. 8e). Furthermore, the cross-sectional area (CSA) analysis of muscle fibers in LLC-control mice is decreased when compared to those of the LLC-muHuR-KO mice (Fig. 8f). Collectively, these results demonstrate that loss of HuR specifically in skeletal muscle protects mice from cancer-induced muscle wasting.

## Discussion

In this work, we demonstrate that, in vivo, the RBP HuR plays an important role in muscle physiology as well as in deciding muscle fate under disease conditions. muHuR-KO mice show a significant increase in exercise endurance, a phenotype that is explained in part by an enrichment of type I fibers. This enrichment most likely arises from an increase in PGC-1α levels, a key regulator of energy metabolism and a promoter of type I muscle fiber formation[4]. These observations establish that, under normal conditions, HuR plays a crucial role in the formation and probably maintenance of type II fibers, and antagonizes the formation of type I fibers, by reducing the expression of PGC-1α. Mechanistically, HuR mediates this effect by forming a complex with the mRNA decay factor KSRP[12], leading to the destabilization of the *PGC-1α* mRNA. Additionally, HuR also participates in the debilitating outcome of cancer cachexia on muscle integrity since the muHuR-KO mice are protected from cancer-induced muscle wasting. Overall, our findings demonstrate that an important function of HuR in adult skeletal muscle is to favor the formation of type II fibers and provide a proof-of-principle that interfering with the function of HuR can prevent cancer-induced muscle loss.

Based on the fact that HuR is required for muscle fiber formation in cell culture[9,11,12], we anticipated that muHuR-KO mice would show strong muscle defects with debilitating consequences. The absence of an obvious and severe phenotype in these mice was, therefore, surprising and unexpected. Interestingly, previous studies have reported similar discrepancy between the ex-vivo (cell culture) and the in vivo impact on the myogenic process of important pro-myogenic factors, such as MyoD and Myf5. While the overexpression of MyoD or Myf5 triggers myogenesis in several cell types, leading to their conversion to muscle fibers[35], the knockout of either one of these genes in mice did not affect muscle development, leading to normal and healthy animals[36,37]. However, the double knockout of both *MyoD* and *Myf5* genes generated mice without functional muscles that die soon after birth[38]. Hence, it was concluded that MyoD and Myf5 proteins have an overlapping role in muscle development and formation during embryogenesis. It is, therefore, possible that this could also be the case for HuR, and functional redundancies with other RBPs could exist to ensure proper muscle fiber formation and function. Indeed, it is well-established that RBPs such as KSRP, Zfp36l1, Zfp36l2, and YB1 (Y-Box-binding protein 1) modulate muscle fiber formation in vitro[13,39,40]. This work and previous observations[12] indicate that HuR collaborates with KSRP to execute some of its function in muscle cells. In addition, similarly to HuR, YB1 promotes muscle fiber formation in vitro, by stabilizing target mRNAs[39]. Hence, it will be of high interest to investigate whether HuR could have functional redundancy with RBPs such as YB1 or others both in vitro and in vivo.

muHuR-KO mice exhibit a significant increase in their exercise endurance capacity, oxygen consumption, and CO2 release. These observations together with high level of oxidative type I fibers in various muscles of muHuR-KO mice, was a clear indication that the depletion of HuR in skeletal muscles is associated with an oxidative phenotype. However, unexpectedly the respiratory exchange ratio (RER) (VCO2/VO2), which is normally used as an indicator of substrate utilization (carbohydrate or fat) by mitochondria for energy production[41], was also increased in the muHuR-KO mice. These results suggest that under conditions of voluntary movement, muHuR-KO mice have an improved rate of carbohydrate oxidation when compared to control littermates. It is well-established, however, that high RER during exercise is associated with reduced endurance in rodent[42], and is a predictor for obesity in human populations[43]. Therefore, the fact that muHuR-KO have a higher RER may seem contradictory to their enhanced exercise endurance. However, this discrepancy could be explained in part by the fact that our calorimetry studies were performed in the absence of imposed exercise conditions. During exercise, substrate utilization for energy production rapidly switches from carbohydrate to fat[42]. Therefore, based on the high endurance level of muHuR-KO mice, we predict that during their engagement into continued physical exercise the RER response will show an increase at the early stages that will be followed by a rapid decrease. This pattern will be consistent with a shift in substrate usage for energy metabolism from carbohydrate to fat as the exercise activity continues and intensifies[41]. Performing such experiments will test this possibility and will shed more light on the role of HuR in muscle function.

Our RNA-seq data indicate that the largest group of genes affected by the depletion of HuR in muscle are those involved in primary metabolic processes such as the PPARα signaling pathway[44]. Interestingly, many of the genes (such as sarcoplasmic reticulum Ca2 + -ATPases (SLN), Ras-related glycolysis inhibitor, and calcium channel regulator (Rrad), insulin receptor substrate 2 (irs2), and peptide transporter 2 (PEPT2) (Supplementary Data 1)) are associated with metabolic imbalance and weight-related diseases[45,46,47]. These results raise the possibility that in vivo HuR could be involved in metabolic plasticity, impacting

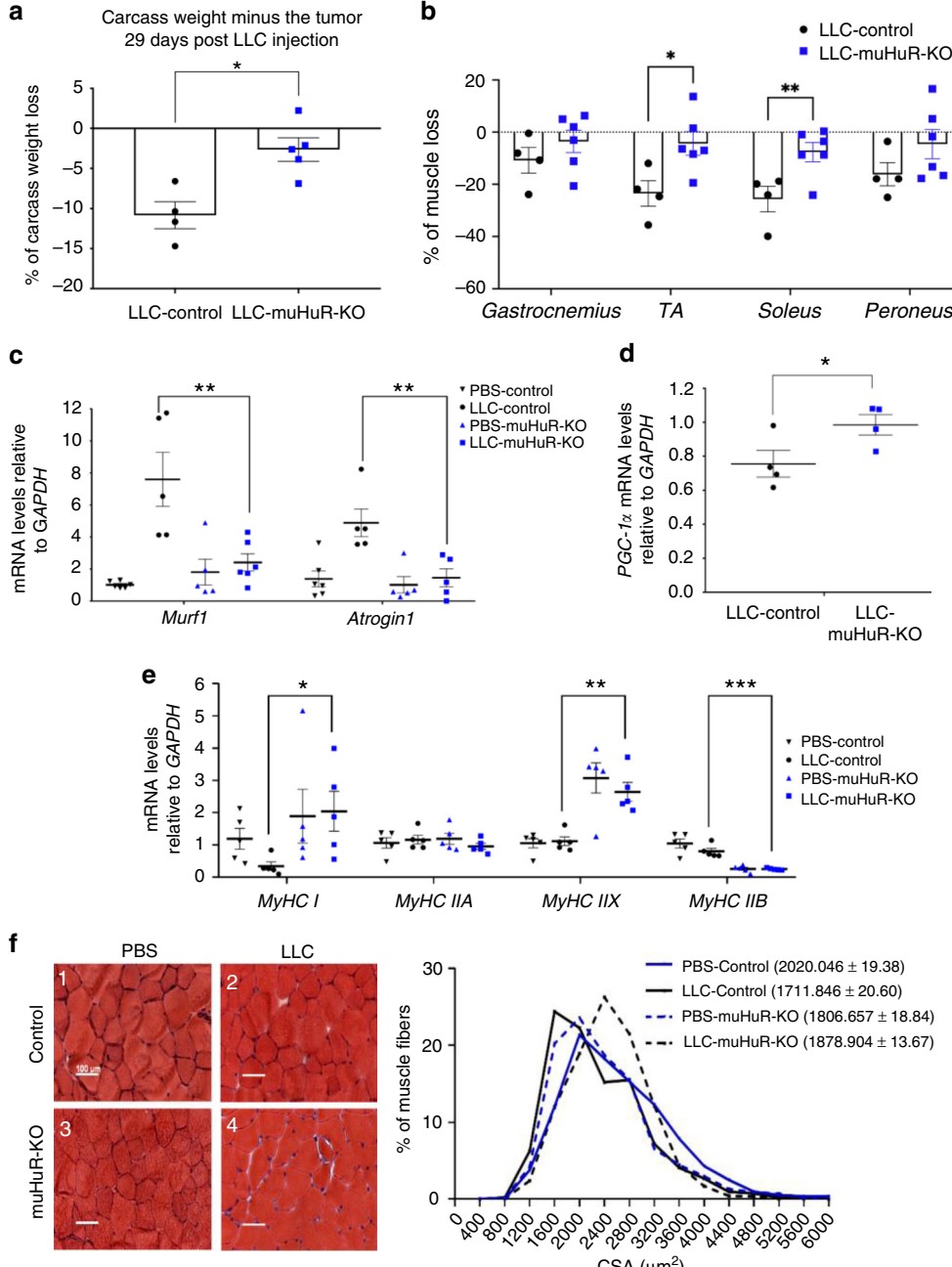

**Fig. 8** HuR ablation in skeletal muscle ameliorates cancer-induced muscle wasting. Muscle atrophy was evaluated in Ctl and muHuR-KO mice using the LLC model of cachexia. **a** Catabolic wasting was assessed post-mortem on tumor-bearing LLC-Ctl and -muHuR-KO mice by measuring total body weight minus tumor weight. The results are presented as mean ± S.E.M *p < 0.05 unpaired t-test, (LLC-Control n = 4, LLC-muHuR-KO n = 5). **b** Muscle atrophy was assessed by determining the relative loss of muscle mass in *gastrocnemius*, TA, *soleus*, and *peroneus* muscles from LLC-Control and LLC-muHuR-KO mice. Percentage (%) of muscle loss in both groups is shown relative to non-tumor PBS injected mice from each cohort (LLC-Control n = 4 and LLC-HuR-KO n = 6). **c–e** Total RNA was isolated from the *gastrocnemius* muscle of control and muHuR-KO mice bearing or not LLC tumors. Relative mRNA expression levels of *Atrogin1 and MuRF1* (**c**), *PGC-1α* (**d**), *MyHC I, MyHC IIA, MyHC IIB, and MyHC IIX* (**e**) was assessed by RT-qPCR and mRNA expression levels was determined relative to *GAPDH* transcript. Expression levels are shown as the fold of induction relative to control-PBS treated mice. (**d**, n = 4), (**c**, **e**, n = 5). **f** Left panel: representative photomicrographs of *gastrocnemius* muscles sections from control and muHuR-KO mice taken after H&E staining. Scale bars = 100 μm. Right panel: frequency histogram showing the distribution of muscle fiber CSA in the *gastrocnemius* muscles from control and muHuR-KO mice bearing or not LLC tumors (n = 4 mice per group). A total of 500 fiber per muscle were used for the CSA analysis. The results are presented as mean ± S.E.M, *p < 0.05, **p < 0.005, ***p < 0.0005 unpaired t-test. Source Data for **a**, **b**, **c**, **d**, **e**, and **f** (right) panels are provided in the Source Data File

muscle function and fate. Future studies will be needed to address the potential involvement of HuR in the onset of metabolic disorders or weight-related diseases.

Our data clearly establish that the enrichment of oxidative type I fibers in muHuR-KO mice is driven, at least in part, by the increased expression of PGC-1α. This observation also indicates that in skeletal muscles HuR promotes a glycolytic type II fibers and that this effect could be mediated by its ability to down-regulate PGC-1α expression by destabilizing *PGC-1α* mRNA in a KSRP-dependent manner. This conclusion is supported by two

facts, (1) our observation that in myoblasts HuR associates with KSRP and that similar to HuR knockdown, the depletion of KSRP stabilizes *PGC-1α* mRNA, (2) an intact HuR/KSRP complex is required for the association of either of these RBPs with *PGC-1α* mRNA. Although, the mRNA destabilizing activity of HuR has been previously reported in several cell lines, including muscle cells[48,12,49]; this effect of HuR was considered a rare event that is either specific to some cell types or is linked to particular growth conditions. The fact that the depletion of HuR in the *soleus* leads to the upregulation of > 85% of the ~1900 affected mRNAs provide a strong indication that in skeletal muscle HuR mainly acts as an mRNA destabilizer rather than a stabilizer. It is likely however, that the impact of HuR on the fate of its mRNA targets in vivo is tissue specific. Indeed, the knockout of HuR specifically in the pyramidal neurons of the hippocampus leads to a significant reduction in the expression levels of the *PGC1α* transcript[50], indicating that in the brain HuR likely acts as a stabilizer for this mRNA. One explanation of this functional dichotomy of HuR is that in different cell types and tissues, HuR associates with various protein ligands and undergoes unique post-translational modifications such as phosphorylation and methylation[9,12,51,52]. However, we still do not know whether any of these regulatory mechanisms affect HuR function in vivo nor their impact on HuR functional switch from stabilizing to destabilizing the same target mRNA.

In addition to the impact of the oxidative phenotype on muscle function and exercise endurance, type I fibers are resistant to atrophy during various disease conditions such as, DMD, denervation, disuse, and cancer cachexia[24,53–55]. In keeping with this, we show that the muHuR-KO mice are protected against cancer (LLC)-induced muscle wasting. Interestingly, despite being composed largely of type I fibers, we observed significant wasting in the *soleus* muscle of wild-type mice, which was prevented in muscles lacking HuR. In fact, the resistance of oxidative muscle fibers to cachectic stimuli is presently under debate, since some reports indicate that type I-rich muscles such as the *soleus* are resistant to cancer-induced muscle wasting, while others have shown the opposite outcome[56,57]. However, while oxidative fibers may or may not be inherently resistant, numerous studies have demonstrated the therapeutic benefit of promoting oxidative metabolic adaptations[24,29,53,54,58]. Since HuR plays a key role in muscle fiber formation in vitro[9,10,12,59,60] and its expression pattern dramatically changes during muscle regeneration in vivo[9], we speculate that HuR could also impact the commitment of satellite cells to myogenesis under cachectic conditions. Indeed, the onset of cachexia-induced muscle wasting is associated with an impairment of the regenerative capacity of satellite cells[61]. Hence, it is possible that under normal conditions HuR promotes satellite cells commitment to the myogenic process, while that under cachectic conditions HuR undergoes a functional switch to become a promoter of muscle loss. Experimentally addressing this possibility would provide more insight into on how HuR could promote both muscle formation and function, as well as the deleterious outcome of cachexia. While more work is needed to understand the mechanisms underlying the atrophic resistance of muHuR-KO mice, our results clearly establish that interfering with HuR function in muscle is protective against muscle atrophy in the LLC model of cancer cachexia.

Our results highlight the possibility that HuR can be considered as a viable target in future strategies to design novel approaches to comabt cancer-induced muscle atrophy. This is of particular interest given the recent development of small molecules that can inhibit HuR function[62–64]. However, given the systemic importance of HuR, future studies using HuR inhibitors to treat muscle atrophy will need to investigate potential side effects, as well as mechanisms for specific delivery to muscle tissue. Some of these issues may be circumvented, however, by targeting HuR functions specifically in muscular tissue. As described above, one apparent unique feature of HuR in muscle is a role in destabilizing target transcripts, which is at least in some cases mediated by interactions with co-factors, such as KSRP. Thus, targeting HuR interaction with protein ligands, such as KSRP, may prove to be a more viable strategy than a direct inhibition of HuR itself.

Overall, our findings are consistent with a model whereby HuR regulates the expression of key modulator of fiber-type specification, thus inhibiting the formation of type I muscle fibers. Taken together, our data suggest HuR is a potential pharmacological target to modulate skeletal muscle metabolism, which could have implications in muscle physiology and diseases such as cancer-induced cachexia.

## Methods

**Animals.** All experiments using animals were approved by the McGill University Faculty of Medicine, Animal Care Committee and comply with guidelines set by the Canadian Council of Animal Care. Mice were housed in a controlled environment and provided commercial laboratory food (Harlan #2018; 18% protein rodent diet; Madison, WI). Housing of the mice was set on a 12 h light–12 h dark cycle. Mice were maintained in sterile cages with corn-cob bedding and had free access to food and water. For our study we used three non-pathogenic mouse strains on a C57BL/6 background; mice expressing cre recombinase under the MyoD promoter (*MyoDCre*)[15], mice in which the exon 2 of the *Elavl1* gene is floxed (*Elavl1fl/fl*)[14], and the HuR muscle-specific knockout mice (*MyoDCre+; Elavl1fl/fl*, muHuR-KO) generated by our laboratory at McGill University using the Cre-LoxP system. The breeding strategy to maintain the muscle-specific HuR knockout colony consisted in back-crossing *Elavl1fl/fl* mice with *MyoDCre+; Elavl1fl/fl* mice. Littermates not expressing Cre recombinase (*MyoDCre-/-Elavl1fl/fl*) were used as control animals in this study.

**Genotyping.** Mouse genomic DNA was isolated in vivo from tail biopsies or ex-vivo from muscle tissue as previously described[65]. Tail biopsies were incubated for 20 min in 300 µl of 0.5 M NaOH at 95 °C. Twenty-five microliters of 1 M Tris-HCl, pH 8 were then added and 2 µl of the resulting sample used as template DNA for PCR amplification of a fragment of the CRE recombinase gene and a fragment containing the LoxP sites. For isolation of genomic DNA from skeletal muscle, tissue was incubated with DNA extraction buffer (100 Mm NaCl, 25 Mm EDTA pH 8, 10 mM Tris-Cl pH 8, 0.5% sodium dodecyl sulfate (SDS), 1 mg/ml Proteinase K) overnight at 56 °C. DNA was then isolated by phenol–chloroform extraction and mice were genotyped by assessing PCR amplification of the Elavl1 gene (exon 2 region). PCR products were visualized by ethidium bromide staining on 2% Agarose gel. Primer sequences are provided in Supplementary Table 2.

**Lewis lung carcinoma (LLC) animal model of cachexia.** Subcutaneous LLC tumors were established in the right hindlimb region of male muHuR-KO mice or control littermates (8–9-weeks-old) by subcutaneously injecting $1 \times 10^6$ LLC cells. During the observation phase (30-day post injection), mice were monitored for tumor size and body weight every other day. The tumor volume was calculated using the formula: $(L \times W^2)/2$, where $L$ is the longest tumor diameter and $W$ the perpendicular axis diameter. At the end of experiment, mice were sacrificed by cervical dislocation and muscles were carefully dissected, weighed, and frozen in liquid nitrogen or in isopentane precooled in liquid nitrogen. The weight of the spleen (an indicator of an active immune response) and hindlimb fat pad was also determined. muHuR-KO mice or control littermates injected with PBS were used as control.

**Energy balance measurement.** Indirect calorimetry study was performed in male, age-matched (8–10-week) muHuR-KO and control mice. Housing was under 12 h light–12 h dark cycle, with free access to food and water. The Oxygen consumption ($\dot{V}O_2$), carbon dioxide production ($\dot{V}CO_2$), and ambulatory activity were measured using a Comprehensive Animal Monitoring System (CLAMS; Columbus Instruments, Columbus, OH) over 72 h, following a 24 h acclimation period. Measurements proceeded under a constant airflow rate of 1000 ml/min. The system allowed for eight individually housed mice to be monitored simultaneously. Oxygen consumption ($\dot{V}O_2$) and carbon dioxide production ($\dot{V}CO_2$) were recorded every 10 min. Respiratory exchange ratio (RER) was calculated as $VO_2/\dot{V}CO_2$. Ambulatory activity was estimated by the number of infrared beam breaks along the x-axis of the metabolic cage. Heat production on per animal basis was calculated from the following equation: $((3.82 + 1.23 \times RER) \times VO_2)$. Data was analyzed using CLAMS examination tool (CLAX; Columbus Instruments) version 2.1.0. Each animal was considered one experimental unit.

**Treadmill exhaustion test**. Male, age-matched (8–10-week) muHuR-KO mice or control littermates were exercised on a Columbus 1050-RM Exer-3/6 Treadmill. The system allowed for six individual mice to be exercised simultaneously. Before the exhaustion test, mice were subjected to an acclimation process for 3 consecutive days with the following program; Day 1: static treadmill band 15 min. Day 2: walking on the treadmill for 15 min (5 m/min). Day 3: running for 10 min (10 m/min). Electric stimulus of 1 Hz was employed to force mice to run. Exhaustion test was conducted on two separate days (2 day resting period in-between) with the following program: 5 m/min for 1 min, 7 m/min for 1 min, 8 m/min for 5 min, followed by an increase of 1 m/min every minute until a maximum velocity of 21 m/min. Exhaustion was consider after 5 s permanence on the electric grid on a 1 Hz, 0.15 mA, 163 V electric stimulus. Maximum exercise capacity was estimated from each run-to-exhaustion trial using three parameters: the duration of the run (min), the distance ran ($m$), and the work performed ($J$). Work was calculated as $W = $ body weight (kg) × running speed (m/min) × running time (min) × grade × 9.8 (J/kg × $m$). Values from the two sessions were averaged to provide exercise capacity. Each animal was considered one experimental unit.

**Rota-rod**. Male, age-matched (20 weeks), muHuR-KO, and control mice were tested on a rota-rod apparatus (Ugo Basile, 47600). Animals received 1 day of training prior to testing with the following program; 5 min on acceleration rotarod at five revolutions per minute (rpm). Exercise days consisted of a program of the following: 2 min at 5 rpm followed by an acceleration period from 5 rpm to 20 rpm in 2 min. Tests were considered finished when mice fell off the apparatus. Exercising sessions (two in all) were completed over the period of a week with 4 days of resting in between the sessions. The latency to fall from the rod was recorded for each trial and values from both sessions were averaged to provide the rotarod latency statistic. If the mouse remained on the rod for more than 30 min, mice were removed from the machine and the test was considered as completed.

**Inverted-grid test**. Fatigability of limbs was tested using the inverted-grid hanging test. Male, age-matched (20 weeks), muHuR-KO, and control animals were placed on a mesh grid (10 × 10 cm) mounted 60 cm above a padded surface. The grid was then inverted, and mouse was suspended upside down. Latency to fall from the grid was recorded; the maximum time allowed per trial was 6 min. Each mouse was tested in two different session consisting of 3 consecutive days each, with 4 days of resting in between sessions. Both sessions were averaged to provide the latency to fall and normalized to total body weight.

**Grip test**. Muscle strength of male muHuR-KO and control mice (20 weeks) was evaluated using a DFE II Series Digital Force Gauge (Ametek DFE II 2-LBF 10-N) with an attached metal grid (10 × 10 cm). Mice were allowed to grasp the metal grid with either two limbs (forelimbs) or four limbs (forelimbs and hindlimbs) and gently pulled along the axis of the grid by the tip of the tail. Maximal strength (Newtons) with which mice pulled the grid was measured in two sessions, each one consisting in triplicate trials over 3 consecutive days. Four days of resting were allowed in-between sessions.

**In situ assessment of muscle contractile function**. To determine contractile function, mice were anesthetized with an intraperitoneal injection of a ketamine-xylazine cocktail (ketamine: 130 mg/kg; xylazine: 20 mg/kg). Anesthesia was maintained with 0.05 ml supplementary doses as needed. The distal tendon of the left TA muscle was isolated and attached in turn with surgical 4.0 silk to the lever arm of a 305C-LR servomotor (Aurora Scientific Instruments), as done previously, with minor modifications (1–3). The Dynamic Muscle Control (DMC) and Analysis Software Suite (Aurora Scientific Instruments) was used for collection and data analysis. The partially exposed muscle surface of the TA was kept moist with PBS (pH 7.4) for the isometric contractile stimulation protocol and was directly stimulated with an electrode placed on the belly of the muscle. In situ measurement of the TA with direct stimulation was chosen over sciatic nerve stimulation, thereby removing potential negative effects such as a central contribution and, because blood delivery is intact, eliminating potential problems of isolated muscles (4). Optimal muscle length and voltage was progressively adjusted to produce maximal tension and length was measured with a microcaliper. The pulse duration was set to 0.2 ms for all tetanic contractions. Force-frequency relationships curves were determined at muscle optimal length at 10, 30, 50, 70, 100, 120, and 150 Hz, with 1 min intervals between stimulations to avoid fatigue. After tetanic-force measurement, the TA muscle was rested for 2 min and then subjected to 60 tetanic contraction. The fatigue resistance protocol was 60 tetanic contraction (75 Hz stimulation/200-ms duration) every 2 s for a total of 2 min. At the end of each experiment, mice were sacrificed by cervical dislocation and muscles were carefully dissected, weighed and frozen in liquid nitrogen or in isopentane precooled in liquid nitrogen. In situ muscle force was normalized to tissue cross-sectional area (expressed as newtons/cm$^2$). Muscle cross-sectional area was estimated by diving muscle mass by the product of the muscle length and muscle density (1.056 g/cm$^3$). During experiments, the investigators were blinded to mice genotype. Statistical analyses for data related to muscle-specific strength were performed using a two-way ANOVA with corrections for multiple comparisons by controlling for the false

discovery rate using the two-stage method of Benjamini and Krieger and Yeku-tieli[66] (with $q < 0.1$ and $p < 0.05$).

**Muscle freezing and sectioning**. *Gastrocnemius*, *soleus*, EDL, and *peroneus* muscles were carefully dissected, mounted on 7% tragacanth gum and snap frozen in liquid-nitrogen-cooled isopentane for 10–20 s. Samples were stored at −80 °C before cryosectioning. Sections (10 µm) were kept at room temperature for 30 min before processing.

**Cross-sectional analysis**. Muscle sections were routinely stained with haema-toxylin and eosin (H&E)[67]. Wide-field images were taken with a 20x objective lens on an inverted Zeiss Axioskop microscope with an Axiocam MRc color camera in the McGill University Life Sciences Complex Advanced BioImaging Facility. Cross-sectional analysis (CSA) of myofibers in muHuR-KO and control mice was determined on muscle sections. Fibers were circled manually, and area determination was calculated using the Image J software (NIH). A minimum of five-hundred fibers per muscle was used for the calculation of the cross-sectional area.

**Immunostaining**. Serial muscle cryosections were incubated with the following monoclonal antibodies from the Developmental Studies Hybridoma Bank: BA-D5 (MyHC-I, 1:500), SC-71 (MyHC-IIA, 1:500), and BF-F3 (MyHC-IIB, 1:500)[68]. The immunohistochemical staining was performed using a goat anti-mouse secondary antibody conjugated to peroxidase-labeled complex (Dako, Glostrup, Denmark). After rinsing in PBS buffer section were incubated for 30 min at room temperature with a freshly prepared solution of 10 mg of 3,3′-diaminobenzidine tetra-hydrochloride (DAB; Sigma, St. Louis, MO) in 15 ml of a 0.05 M Tris buffer at pH 7.6, containing 1.5 ml of 0.3% $H_2O_2$. Sections were then analyzed using a 20 x objective lens on an inverted Zeiss Axioskop microscope.

**Cell culture**. Murine Lewis Lung carcinoma cells (LLC) were obtained from the ATCC and grown in Dulbecco's modified eagle medium (DMEM) with 10% FBS and 1% streptomycin–penicillin (Invitrogen). C2C12 myoblasts (ATCC, Manassas, VA, USA) were grown and maintained in DMEM (Invitrogen) containing 20% Fetal Bovine Serum (FBS), and 1% penicillin/streptomycin antibiotics (Invitrogen). All cells were grown in a humidified incubator at 37 °C, 5% $CO_2$.

**Transfection**. The transfection of siRNA into C2C12 cells was performed as previously described[12]. Briefly, the transfection with siHuR, siKSRP or siCtl was performed when cells were 20–30% confluent. The transfection treatment was repeated 24 h later when cells were 50–60% confluent. All siRNAs duplexes were used at a final concentration of 120 nM. The GFP and GFP-HuR plasmids were generated and used as described in ref. [9]. *jetPRIME*® (Polyplus) transfection regent was used for all transfections following the manufacturer's instructions. siRNA oligonucleotides against siHuR, siKSRP as well as the control siRNA Ctl, was obtained from Dharmacon. siRNA sequences are provided in Supplementary Table 2.

**Preparation of muscle/cell extracts and immunoblotting**. Muscle extracts were prepared by homogenization of frozen muscle tissue in extraction buffer (1x PBS,1% NP-40, 0.5% DOC, 0.1% SDS, 2 mM SOV, 1x protease inhibitor (Roche)). Cell extracts were prepared by incubating C2C12 muscle cells with lysis buffer (50 mM HEPES pH 7.0, 150 mM NaCl, 10% glycerol, 1% Triton, 10 mM pyropho-sphate sodium, 100 mM NaF, 1 mM EGTA, 1,5 mM MgCl2, 1x protease inhibitor (Roche) for 15 min on ice. The lysed muscle/cells were then centrifuged at 0.1 times gravity ( x g) 12,000 rpm for 15 min at 4 °C in order to collect the supernatant. Western blot experiments were performed as previously described in ref. [12]. Western blots were probed with antibodies against HuR (3A2, 1:10,000), PGC-1α (abcam, 1:1000), KSRP (abcam, 1:5000), Oxphos Antibody Cocktail (containing five mouse antibodies against the CI subunit NDUFB8, CII-30kDa, CIII-Core protein 2, CIV subunit I, and CV alpha subunit (abcam, 1:1000) or α-tubulin as loading control (Developmental studies Hybridoma Bank, 1:1000).

**Reverse transcription quantitative PCR (RT-qPCR)**. Total RNA was isolated using TRIzol reagent (Invitrogen). One microgram of total RNA was reverse transcribed using the M-MuLV RT system according to the manufacturer's instructions (New England BioLab). A 1/80 dilution of complementary DNA was then used to assess mRNAs expression using SsoFast™ EvaGreen® Supermix (Bio-Rad). Expression level of genes of interest were standardized using *GAPDH* or *RPL32* as reference, and relative levels of expression were quantified by calculating $2^{-\Delta\Delta C_T}$, where $\Delta C_T$ is the difference in $C_T$ (cycle number at which the amount of amplified target reaches a fixed threshold) between target and reference genes. Primer sequences can be found in Supplementary Table 2.

**Actinomycin D pulse-chase experiments**. The stability of the mRNAs of interest was assessed by the addition of the RNA polymerase II inhibitor, actinomycin D (ActD) (5 µg/ml) to GFP, GFP-HuR, siHuR, siKSRP, and siCtl-treated cells over a

6 h period. Total RNA was isolated at the indicated time points, using TRIzol reagent (Invitrogen), and analyzed by RT-qPCR. The expression level of the different mRNAs at each time point was determined relative to *RPL32* mRNA levels and plotted relative to the abundance of each message at 0 h of ActD treatment that is considered as 1.

**Polysome fractionation**. Polysome fractionation was performed as previously described[12]. Briefly, the cytoplasmic extracts obtained from myoblasts treated with or without siHuR, collected at 100% confluency, were centrifuged at 130,000 × *g* for 2 h on a sucrose gradient (15–50% w/v). Absorbance at wavelength 254 nm was measured in order to determine the profile of polysome distribution. Twenty fractions were collected and divided in two groups; non-polysome (NP, fractions 1–6) and polysome (P, fractions 7–20). RNA was then extracted from each group using TRIzol LS (Invitrogen) according to the manufacturer's instructions. RNA integrity was monitored on agarose gel and was then analyzed by RT-qPCR using specific primers for PGC-1α and GAPDH mRNAs. *PGC-1α* mRNA level was standardized against GAPDH mRNA in each group and plotted as a Polysome to Non-Polysome ratio.

**Immunoprecipitation/RNA-IP**. Immunoprecipitation/RNA-IP experiments were performed as previously described[12]. Briefly, 15 μl of the anti-HuR, anti-KSRP, or IgG antibodies were incubated with 60 μl of protein A-Sepharose slurry beads (washed and equilibrated in cell lysis buffer) for 4 h at 4 °C. Beads were washed three times with cell lysis buffer and incubated with 500 μg of cell extracts overnight at 4 °C. Beads were then washed again three times with cell lysis buffer and co-immunoprecipitated RNA was then eluted and processed for RT-qPCR analysis.

**mRNA sequencing (RNA-seq)**. Total RNA from *soleus* muscle from two muHuR-KO and two control mice was isolated using TRIzol reagent (Invitrogen). RNA samples were assessed for quantity and quality using a NanoDrop UV spectrophotometer (Thermo Fisher Scientific Inc), and a Bioanalyser (Agilent Technology Inc). The four RNA-seq libraries were sequenced on the Illumina NextSeq 500 platform at the Institute for Research in Immunology and Cancer (IRIC) Genomics Core Facility, University of Montreal, to produce over 60 million, 100 nucleotide paired-end reads per sample. The reads were then trimmed for sequencing adapters and aligned to the reference mouse genome version mm10 (GRCm38) using Tophat version 2.0.10. Gene quantification was performed on the mapped sequences using the htseq-count software version 0.6.1. We performed a differential expression analysis using DESeq2 package, log transformation was used to normalize raw read counts and to calculate normalized expression counts. Biological replicates were combined, and the dataset visualized on a heatmap using the Morpheus software (version 4.7). (Data can be accessed in GEO database, GSE134241).

**Ingenuity pathway analysis (IPA)**. The RNA-seq dataset generated by DESeq analysis was subjected to a subsequent analysis using the "Ingenuity Pathways Analysis" Software 5.0 (IPA, Ingenuity Systems) to define the main biologic processes associated with the gene expression changes in the muHuR-KO mice. This analysis was performed using detectably expressed (read number > 0) genes across all samples (https://www.qiagenbioinformatics.com/products/ingenuitypathway-analysis).

**Mitochondria number**. Mitochondrial number was estimated by determining the mitochondrial to nuclear DNA ratio as previously described[69]. Briefly, genomic DNA from skeletal muscle tissue was incubated with DNA extraction buffer (100 Mm NaCl, 25 Mm EDTA pH 8, 10 mM Tris-Cl pH 8, 0.5% SDS, 1 mg/ml Proteinase K) overnight at 56 °C. DNA was then isolated by phenol–chloroform extraction. DNA was quantified and diluted to a final concentration of 10 ng/μl to be used for qPCR amplification. A comparison of ND1 (NADH dehydrogenase 1, mitochondrial gene) expression relative to HK2 (Hexokinase 2, nuclear gene) DNA expression was used to estimate mtDNA copy number to nDNA copy number ratio.

**Statistical analyses**. All values are reported as mean ± standard error of the mean (S.E.M). Significant differences between two group means were discerned by unpaired *t*-tests for normally distributed variables. Normality determined using the D'Agostino–Pearson test were appropriate. Statistical analyses for in-situ consisted in two-way ANOVA with corrections for multiple comparisons by controlling for the false discovery rate using the two-stage method of Benjamini and Krieger and Yekutieli[66]. *p*-values of < 0.05 were considered significant.

**Reporting summary**. Further information on research design is available in the Nature Research Reporting Summary linked to this article.

## Data availability
The data reported in this study in support of all the findings outlined are available from the corresponding author upon reasonable request. The source data underlying Figs. 1–8

and Supplementary Figs. 1–4 and 6–10 are provided as a Source Data file. The raw RNASeq data have been deposited into NCBI Gene Expression Omnibus (GEO) database under accession number GSE134241.

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

## Acknowledgements

We are grateful to Xian Jin Lian for technical help with some of the experiments in the manuscript and Amr Omer for reading and commenting on the manuscript. We thank Michel Tremblay and Maxime Bouchard for their ideas, suggestions, and guidance. We thank Dr. Siegfried Hekimi and Mrs. Eve Bigras for their help with the grip test experiment. This work is funded by CIHR operating grants (MOP-142399, MOP-89798), and a NSERC Discovery grant RGPIN-2014–06035 to I.E.G. B.J.S was funded by a scholarship received from the Concejo Nacional de Ciencia y Tecnologia (CONACyT), the Fonds de recherche du Québec— Nature et technologies (FRQNT) and the Biochemistry department at McGill University. D.T.H. was funded by a scholarship received from the Canadian Institute of Health Research (CIHR) funded Chemical Biology Program at McGill University. J.F.M. was supported by the CIHR/FRSQ training grant in cancer research of the McGill Integrated Cancer Research Training Program.

## Author contributions

B.J.S. performed, analyzed and helped in the interpretation of all the experiments in the manuscript and wrote the original draft. A.K.T. assisted with sample preparation as well as acquisition and analysis of data in the majority of the in vivo and in vitro experiments. D.T.H. contributed to the investigations and validations of the cachectic experiments and helped edit the manuscript. P.L.H. provided technical expertize and help in sample preparation and data acquisition for Fig. 4a and Supplementary Figs. 3 and 4. S.M., J.M. and B.L.P. assisted in sample preparation and data acquisition of the mRNA stability experiments. S.D.-M. assisted with the conceptualization, data analysis, and helped edit and review the manuscript. E.K and D.K assisted in the generation of muHuR-KO mice. J.P.L.G. and S.N.A.H. designed and performed the in situ analysis experiments and helped in interpreting and describing the data. A.H.C. and K.H. discussed the data, reviewed and helped edit the manuscript. I.-E.G. conceptualized, established, and directed the execution of research goals, interpreted the data, reviewed, and edited the manuscript.

## Additional information

**Competing interests:** The authors declare no competing interests.

