## [Peer Review File · Nature Communications]

Reviewers' Comments:

Reviewer #1:

Remarks to the Author:

In this study, Sanchez et al. characterize the muscle specific HuR knock-out mouse (muHuR-KO) to investigate the in vivo role of HuR in muscle. Despite having a known role in muscle differentiation of cultured muscle cells, muHuR-KO mice did not have any overt muscle defects. The authors found that HuR-deficient mice exhibited enhanced exercise endurance and increased muscle oxidative capacity. Accordingly, HuR-deficient muscles contained predominantly type I oxidative myofibers. By RNA-seq comparative analysis of control and muHuR-KO muscles, the authors identify an upregulation in the PPARalpha/PGC-1alpha pathway. Mechanistically, HuR normally destabilizes PGC-1alpha mRNA, which is suggested to occur via the KH-type splicing regulatory protein (KSRP). Lastly, the authors induced cachexia in the muHuR-KO mice and found that mice lacking HuR in muscle are resistant to cancer-induced muscle wasting.

While the role of PGC-1alpha in type I oxidative myofiber formation and its ability to protect against muscle wasting are not novel findings, the implication of HuR as a target for the treatment of cachexia is very interesting and important. A few key experiments (outlined below) are required for a more complete understanding of how HuR is important for muscle function and muscle wasting.

Major comments:

1. Due to the lack of an obvious muscle phenotype in the muHuR-KO animals, the authors propose that HuR-mediated myogenic functions are more relevant in post-natal muscle development. Since loss of HuR blocks differentiation of cultured muscle cells in vitro, a very simple experiment to assess if HuR is important for postnatal myogenic differentiation is to perform a muscle regeneration experiment in the KO animals via muscle injury methods and assess muscle regeneration capacity.
2. The authors performed a treadmill exhaustion test and limb grip strength assay to assess muscle function. While these tests are non-invasive and do assess muscle function in vivo, ex vivo muscle force measurements give more specific read outs and can provide more information (muscle force, velocity, power, fatigue, etc). Ex vivo muscle force measurements would complement the in vivo data and provide a more complete characterization of HuR's role in muscle.
3. In Figure 4, fiber type is shown as a percentage of total muscle fibers. The authors should also enumerate total fiber number in control vs. muHuR-KO animals.
4. In the RNA-seq analysis performed, 1914 genes were affected (86% were increased and 14% were decreased). Were genes already known to be regulated by HuR reflected in this list? Was MyoD among the 14% of genes that are decreased? This data should be reflected as a figure.
5. From the results presented in Figures 6D-G, it is unclear if HuR is working together with KSRP to mediate destabilization of PGC-1alpha mRNA. Does KSRP still bind to PGC-1alpha mRNA in the absence of HuR? This experiment could easily be performed by siRNA knock-down of HuR in C2C12 cells.
6. Does over-expression of HuR result in reduced steady state levels and mRNA half-life of PGC-1alpha?
7. A key finding in this study is that the loss of HuR in muscle is protective against muscle wasting in the LLC model of cancer cachexia. This study would be enhanced if this protective effect can be recapitulated using small molecule inhibitors of HuR. Would pre-treatment of the LLC cells with HuR inhibitors prior to injection into the mice prevent muscle atrophy? Or would direct intra-

muscular injections of HuR inhibitors into cachectic muscles ameliorate muscle wasting?

Minor comments:

1. On page 6, paragraph 1, the last sentence reads "The knockout of HuR is initiated in satellite cells during embryogenesis, since Cre under the MyoD promoter is activated in branchial arches and limb buds as early as day E10". By definition, these MyoD+ cells are not satellite cells. This needs to be rephrased and changed to either myogenic or muscle progenitor cells.

2. Both in the Results and Discussion sections, the authors mention that other RNA binding proteins may be compensating for the loss of HuR. What other RBPs have been implicated in myogenesis? The authors could speculate on potential candidates that have functional redundancy with HuR.

Reviewer #2:

Remarks to the Author:

The corresponding author is a productive expert in several areas of muscle metabolism including cancer induced muscle wasting, and has a long-standing interest in HuR regulation, which has not been widely examined in muscle with cancer cachexia or exercise. The study examine muscle specific HuR loss and characterized the phenotype.. The authors report HuR loss protects against increases muscle endurance, cancer induced muscle wasting and promotes a type 1 phenotype.

Specific

A strong premise is provided for the importance of fiber type, cellular mechanisms related to fiber type, and the importance of improving the understanding of HuR regulation . The study performs an impressive number of in vivo functional tests and measurements.

Overall the muscle measurements describe increased whole-body endurance and decreased wasting susceptibility in the HuR KO. While an interesting discovery analysis on muscle gene expression is included, the analysis of muscle related to the specific functional endurance change is superficial, as is the analysis of muscle wasting. The strongest data is in the overall characterization of the knockout phenotype. The study impact would be strengthened by more detail in muscle related to the impact of HuR on oxidative metabolism, which is inferred by myosin expression. Additional markers of oxidative enzyme capacity, muscle uncoupling expression would improve the study in the absence of specific muscle function and mitochondrial function.

Several concerns were identified with rigor and reproducibility. The statistical approaches need described. The N for experiments with only 3 or 4 observations needed bolstered . The replication and repeatability strategy of cell culture experiments needs addressed, and details on sampling and quantification or histological and immunohistochemical analysis of muscle needs improved. Also, page 7 and elsewhere, there are not "non-significant decreases", it is only decreased if the statistical analysis established the change. It is also unclear how controls in Figure 4c have no variation. The controls should still be 1.0 +/- the s.e.m. Figure 5c lacks statistical symbols. Only an N=2 was used for fiber CSA analysis in figure 7 and needs to be increased.

Do the authors expect the use of MyoD as the muscle specific promoter had any affect on satellite activity? Would this alter any interpretations. Do the authors have data.

Abstract, the interpretation of the results is far to strong as the study provides no direct evidence that a fiber type shift protected the KO muscle from wasting. They were just associated. Please edit.

Introduction and Abstract: A we defined purpose centered on the main variables in the study with

and associated hypothesis would improve the impact of the study, as there is currently an open-ended discovery approach which makes the study look much more like a phenotype description after KO. From the mechanisms nicely described in the introduction it looks as if this type of statement could be formulated.

Introduction: While comprehensive, paragraphs 1 and 2 have redundancy in concept and should be combined.

Methods page 20. Please provide the minimum number of muscle fibers quantified per muscle to determine CSA

Methods cell culture. Please specify how long C2C12 myoblasts were allowed to differentiate into myotubes.

Statistical analysis was not included in the methods or figure legends.

Reviewer #3:

Remarks to the Author:

Sánchez and colleagues have studied how skeletal muscle-specific ablation of the HuR gene, encoding an RNA-binding protein, affects muscle fiber type distribution and function. Of particular interest are the discovery of HuR-dependent destabilization of PGC-1 α mRNA, and a protective effect of muscle-specific HuR knockout in a mouse model for cancer cachexia. Even though the study shows convincing data and results in regards to the contractile fiber type, it might be further improved by considering the following aspects:

- 1.) The metabolic phenotype, which can differ from the myofibrillar fiber type, should be characterized: no data about fatty acid oxidation, mitochondrial and other metabolic genes/proteins/activities are currently included. Similarly, it would be interesting to see whether the myofibrillar fiber type shift is associated with a modulation of mitochondrial number, activity and/or morphology.
- 2.) The authors should show RER data: not only would these allow a conclusion about substrate usage, but would also explain why the “heat production” results (which do not really provide a measure of body temperature!) are unchanged despite the consistent increase in VO₂.
- 3.) Based on the hypothesis put forward in this manuscript, a higher expression of HuR would be expected in glycolytic fibers. However, figure 1, panel F suggests increased HuR expression in soleus. How do the authors reconcile these findings?
- 4.) In the cachexia model, expression of PGC-1 α should be determined to link the proposed mechanism to the reported outcome.
- 5.) The authors claim that increased expression of PPAR α is linked to the enrichment of type I fibers in skeletal muscle. However, muscle-specific overexpression of PPAR α results in more glycolytic fibers, whereas muscle-specific deletion of PPAR α induces an oxidative shift (see Gan et al. 2013 J Clin Invest 123: 2564–2575). The role of PPAR α in this context thus is much more controversial compared to other important regulators, including PPAR δ , ERR α or ERR γ , that are clearly associated with oxidative muscles. The authors should measure the expression of these and related genes in their paradigm.
- 6.) Figure 1: Tubulin levels seem higher in HuR KO animals compared to controls?
- 7.) Figure 2, panel F: is the difference statistically significant as claimed in the text?

8.) Why was the number of type IIX not quantified? These would be the unstained fibers.

9.) The authors claim that “the upregulation of PGC-1alpha is associated [...] with the prevention of muscle atrophy induced by several diseases including cancer.” I am not aware of any data showing this. In fact, this seems not to be the case, and PGC-1alpha overexpression as been linked to increased tumor size (see Wang et al. 2012 PLoS ONE 7(3):e33426).

10.) Data from the soleus muscle have been included in the main text (Figures 4 and 5), while the fiber size distribution of other muscles are shown in the supplement, whereas gene expression data of these muscles remain elusive. Even though effects are apparent in the soleus, this muscle might not represent the best model for the experimental context brought forward in this manuscript, since this muscle already is highly oxidative. Thus, even stronger effects of HuR deletion would be expected in mixed and glycolytic muscles, which have a higher window for a shift towards an oxidative phenotype. For example, the absence of changes in the expression of most of the markers determined by qPCR in Figure 5 might be due to the already high basal levels of these in soleus. It thus would be interesting to see analogous data from other muscles.

Point-by-point rebuttal to reviewers' comments

We would like to thank the reviewers for the great job evaluating our manuscript and for the constructive comments and suggestions, the response to which have significantly strengthened our data and message. We addressed all reviewers' comments both experimentally and by amending the text.

In addition to the fully assembled manuscript, we are also providing a copy of the full manuscripts in which we highlight the changes made. These changes are highlighted in Bold Yellow and explanation are provided on the side using track changes.

Below is a copy of the reviewer reports and our detailed answers. The reviewer comments are in normal font, and are followed immediately by our response, highlighted in grey.

Reviewer #1 (Remarks to the Author):

In this study, Sanchez et al. characterize the muscle specific HuR knock-out mouse (muHuR-KO) to investigate the in vivo role of HuR in muscle. Despite having a known role in muscle differentiation of cultured muscle cells, muHuR-KO mice did not have any overt muscle defects. The authors found that HuR-deficient mice exhibited enhanced exercise endurance and increased muscle oxidative capacity. Accordingly, HuR-deficient muscles contained predominantly type I oxidative myofibers. By RNA-seq comparative analysis of control and muHuR-KO muscles, the authors identify an upregulation in the PPARalpha/PGC-1alpha pathway. Mechanistically, HuR normally destabilizes PGC-1alpha mRNA, which is suggested to occur via the KH-type splicing regulatory protein (KSRP). Lastly, the authors induced cachexia in the muHuR-KO mice and found that mice lacking HuR in muscle are resistant to cancer-induced muscle wasting.

While the role of PGC-1alpha in type I oxidative myofiber formation and its ability to protect against muscle wasting are not novel findings, the implication of HuR as a target for the treatment of cachexia is very interesting and important. A few key experiments (outlined below) are required for a more complete understanding of how HuR is important for muscle function and muscle wasting.

We thank the reviewer for the kind words and encouraging comments on the impact of our study.

Major comments:

1. Due to the lack of an obvious muscle phenotype in the muHuR-KO animals, the authors propose that HuR-mediated myogenic functions are more relevant in post-natal muscle development. Since loss of HuR blocks differentiation of cultured muscle cells in vitro, a very simple experiments to assess if HuR is important for postnatal myogenic differentiation is to perform a muscle regeneration experiment in the KO animals via muscle injury methods and assess muscle regeneration capacity.

[Redacted]

2. The authors performed a treadmill exhaustion test and limb grip strength assay to assess muscle function. While these tests are non-invasive and do assess muscle function *in vivo*, *ex vivo* muscle force measurements give more specific read outs and can provide more information (muscle force, velocity, power, fatigue, etc). *Ex vivo* muscle force measurements would complement the *in vivo* data and provide a more complete characterization of HuR's role in muscle.

We are grateful to the reviewer for this excellent suggestion and agree that such an experiment “complements the *in vivo* data and provides a more complete characterization of HuR's role in muscle”. We performed *in situ* Analysis of Force and Fatigability that are now included as **Figure 2A**.

3. In Figure 4, fiber type is shown as a percentage of total muscle fibers. The authors should also enumerate total fiber number in control vs. muHuR-KO animals.

We agree with this suggestion and now the total fiber number in control vs. muHuR-KO animals is included in Figure 4 of this revised manuscript and shown as **Figure 4C**. In addition, we are also providing a table outlining the total fiber number of Soleus, EDL, and Peroneus muscles (**Supplemental Table 1** of the revised manuscript).

4. In the RNA-seq analysis performed, 1914 genes were affected (86% were increased and 14% were decreased). Were genes already known to be regulated by HuR reflected in this list? Was MyoD among the 14% of genes that are decreased? This data should be reflected as a figure.

Yes, several known HuR target genes were indeed in this list. For example, as expected MyoD (Van der Giessen et al., JBC 2003) was among the 14% transcripts that were decreased in muHuR-KO muscle. On the other hand, and as expected (Cammis et al., Nature Communications 2014), nucleophosmin (NPM) mRNA was among the 86% genes that were up-regulated in muHuR-KO muscle. These data are now included as Figure 5B in the revised manuscript. In this figure, we also show, that HuR depletion does not affect the expression levels of HMGB1 mRNA, consistent with our previous observations (Dormy-Raclet et al., Nature Communications 2013) where HuR impacts the translation of this transcript but not its half-life.

5. From the results presented in Figures 6D-G, it is unclear if HuR is working together with KSRP to mediate destabilization of PGC-1alpha mRNA. Does KSRP still bind to PGC-1alpha mRNA in the absence of HuR? This experiment could easily be performed by siRNA knock-down of HuR in C2C12 cells.

This is a great suggestion. The experiment was performed, and the results are included in the revised manuscript as part of a new **Figure 7**. We show that the binding of KSRP to PGC-1 α mRNA depends on HuR expression and the reverse is also true (HuR binding to PGC-1 α mRNA requires KSRP expression).

6. Does over-expression of HuR result in reduced steady state levels and mRNA half-life of PGC-1alpha?

We are grateful for this question. The answer is yes. Overexpressing HuR in muscle cells significantly reduced the expression levels of PGC-1 α protein as well as the steady state levels and half-life of its mRNA. These data are included as part of **Figure 6** in the revised manuscript and shown as **Figures 6F-I**.

7. A key finding in this study is that the loss of HuR in muscle is protective against muscle wasting in the LLC model of cancer cachexia. This study would be enhanced if this protective effect can be recapitulated using small molecule inhibitors of HuR. Would pre-treatment of the LLC cells with HuR inhibitors prior to injection into the mice prevent muscle atrophy? Or would direct intra-muscular injections of HuR inhibitors into cachectic muscles ameliorate muscle wasting?

[Redacted]

Minor comments:

1. On page 6, paragraph 1, the last sentence reads “The knockout of HuR is initiated in satellite cells during embryogenesis, since Cre under the MyoD promoter is activated in branchial arches and limb buds as early as day E10”. By definition, these MyoD+ cells are not satellite cells. This needs to be rephrased and changed to either myogenic or muscle progenitor cells.

We would like to thank the reviewer for catching this oversight. The statement was changed to include “muscle progenitor cells”.

2. Both in the Results and Discussion sections, the authors mention that other RNA binding proteins may be compensating for the loss of HuR. What other RBPs have been implicated in myogenesis? The authors could speculate on potential candidates that have functional redundancy with HuR.

This is a great point and the discussion was amended to include a whole new paragraph that provides examples of other RNPs that could have redundant function with HuR in muscle cells/fibers (see second paragraph of the discussion of the revised manuscript). We also discuss future avenue to test this redundancy.

Reviewer #2 (Remarks to the Author):

The corresponding author is a productive expert in several areas of muscle metabolism including cancer induced muscle wasting, and has a long-standing interest in HuR regulation, which has not been widely examined in muscle with cancer cachexia or exercise. The study examine muscle specific HuR loss and characterized the phenotype. The authors report HuR loss protects against increases muscle endurance, cancer induced muscle wasting and promotes a type 1 phenotype.

Specific

- A strong premise is provided for the importance of fiber type, cellular mechanisms related to fiber type, and the importance of improving the understanding of HuR regulation. The study performs an impressive number of in vivo functional tests and measurements.

We thank the reviewer for the nice comments on the work done in this study.

- Overall the muscle measurements describe increased whole-body endurance and decreased wasting susceptibility in the HuR KO. While an interesting discovery analysis on muscle gene expression is included, the analysis of muscle related to the specific functional endurance change is superficial, as is the analysis of muscle wasting. The strongest data is in the overall characterization of the knockout phenotype. The study impact would be strengthened by more detail in muscle related to the impact of HuR on oxidative metabolism, which is inferred by myosin expression. Additional markers of oxidative enzyme capacity, muscle uncoupling expression would improve the study in the absence of specific muscle function and mitochondrial function.

We agree with the reviewer that the addition of oxidative markers will strengthen the study. We included: Force and Fatigability analysis (**Figure 2A of the revised manuscript**) to further study muscle function. We assessed the stability of the electron transport chain (ETC) and, consistent with our oxidative phenotype, we observed an increase in the levels of CIII and CIV as well as the ATP synthase 5A (**Figure 3E of the revised manuscript**). We also measured other markers such as PPAR α (**Figure 5D, Supplemental Figure 6 of the revised manuscript**), Acadv1, CD36, FAS, LDL, UCP-2, UCP-3, NRF1, NRF2 and mDNA/nuDNA (**Supplemental Figure 7 of the revised manuscript**). These measurements show an increase in NRF-1 expression, which is one of the main effectors of PGC-1 α -induced mitochondria function. However, we did not see any major changes in other markers, including mitochondria number.

- Several concerns were identified with rigor and reproducibility.

All the identified shortcomings were addressed as described below

- The statistical approaches need to be described.

A clear description of the statistical analysis is now included in a separate paragraph in the Material and Method section as well as in legends of each figure, where appropriate.

- The N for experiments with only 3 or 4 observations needed bolstered. The replication and repeatability strategy of cell culture experiments needs addressed, and details on sampling and quantification or histological and immunohistochemical analysis of muscle needs improved.

The N of the majority of our cell culture, western blots and animal experiments has been increased as requested. Also, more details on the sampling and quantification of our experiments has been included in both the Materials and Method section and the Figure Legends.

- Also, page 7 and elsewhere, there are not “non-significant decreases”, it is only decreased if the statistical analysis established the change.

We thank the reviewer for raising this point. The text throughout the manuscript has been corrected accordingly.

- It is also unclear how controls in Figure 4c have no variation. The controls should still be 1.0 +/- the s.e.m.

Thank you so much for pointing out this issue and we apologize for this oversight. This has been corrected and the figure showing variation in the control is now included as **Figure 4D** in the revised manuscript.

- Figure 5c lacks statistical symbols.

The symbol has been added where significance was observed.

- Only an N=2 was used for fiber CSA analysis in figure 7 and needs to be increased.

This issue has been corrected and the N has been increased. These data are now part of the **new Figure 8** in the revised manuscript.

- Do the authors expect the use of MyoD as the muscle specific promoter had any affect on satellite activity? Would this alter any interpretations. Do the authors have data.

This is a great point. As stated in our answer to point 1 of reviewer 1, the regeneration capacity of muHuR-KO mice is similar to that of the control mice. In our opinion, the reason behind this lack of effect on regeneration could be due to the fact that MyoD is expressed late in the process and muscle progenitor cells are already competent for differentiation when HuR is removed by MyoD-Cre. We hope that our future experiments using earlier drivers, such as Myf5 and Pax7 promoters will give us a clearer answer.

- Abstract, the interpretation of the results is far too strong as the study provides no direct evidence that a fiber type shift protected the KO muscle from wasting. They were just associated. Please edit.

Point well-taken and correction has been made as suggested.

- **Introduction and Abstract:** A well-defined purpose centered on the main variables in the study with and associated hypothesis would improve the impact of the study, as there is currently an open-ended discovery approach which makes the study look much more like a phenotype description after KO. From the mechanisms nicely described in the introduction it looks as if this type of statement could be formulated.

We really appreciate this suggestion and we completely agree. The text of the introduction in the revised manuscript has been amended to reflect the point raised (see paragraph 4 of the introduction of the revised manuscript).

- **Introduction:** While comprehensive, paragraphs 1 and 2 have redundancy in concept and should combined.

We agree, and both paragraphs were combined.

- **Methods page 20.** Please provide the minimum number of muscle fibers quantified per muscle to determine CSA.

The minimum number of muscle fibers quantified per muscle to determine CSA has been included in the Method section as requested

- **Methods cell culture.** Please specify how long C2C12 myoblasts were allowed to differentiate into myotubes.

As we have previously shown, (Vander Geissen et al., JBC 2003; Beauchamps et al. CDD 2010; Dormoy_Raclet et al., Nature Communications 2013; etc.) in the absence of HuR, C2C12 cells do not enter the myogenic process. Therefore, C2C12 were used at confluency at Day 1 of the differentiation process. This information has been included where appropriate.

- Statistical analysis was not included in the methods or figure legends.

Statistical analysis are now included in all figure legends and method sections.

Reviewer #3 (Remarks to the Author):

Sánchez and colleagues have studied how skeletal muscle-specific ablation of the HuR gene, encoding an RNA-binding protein, affects muscle fiber type distribution and function. Of particular interest are the discovery of HuR-dependent destabilization of PGC-1 α mRNA, and a protective effect of muscle-specific HuR knockout in a mouse model for cancer cachexia. Even though the study shows convincing data and results in regards to the contractile fiber type, it might be further improved by considering the following aspects:

1) The metabolic phenotype, which can differ from the myofibrillar fiber type, should be characterized: no data about fatty acid oxidation, mitochondrial and other metabolic genes/proteins/activities are currently included. Similarly, it would be interesting to see whether the myofibrillar fiber type shift is associated with a modulation of mitochondrial number, activity and/or morphology.

We thank the reviewer for raising this important point and, as per our answer to point one of reviewer 2, we agree that including such studies strengthens our paper. We assessed the stability of the electron transport chain (ETC) and, consistent with our oxidative phenotype, we observed an increase in the levels of CIII and CIV as well as the ATP synthase 5A (**Figure 3E of the revised manuscript**). We also measured other markers such as PPAR α (Figure 5D, Supplemental Figure 6 of the revised manuscript), Acadv1, CD36, FAS, LDL, UCP-2, UCP-3, NRF1, NRF2 and mtDNA/nuDNA (Supplemental Figure 7 of the revised manuscript). These measurements show an increase in NRF-1 expression, which is one of the main effectors of PGC-1 α -induced mitochondrial biogenesis and function. However, we did not see any major changes in other markers, including mitochondria number.

2) The authors should show RER data: not only would these allow a conclusion about substrate usage, but would also explain why the “heat production” results (which do not really provide a measure of body temperature!) are unchanged despite the consistent increase in VO₂.

This is an excellent suggestion and we totally agree with. As requested, we performed this experiment, measured RER, and the data were somewhat surprising. Under our experimental conditions (resting and voluntary movement) muHuR-KO mice exhibited a slight increase in RER, indicating that, under these conditions, these mice preferentially use carbohydrate as a source of energy. These data are included as **Figure 3D in the revised manuscript**. We included a new paragraph in the discussion section to highlight these data (third paragraph in the discussion section of the revised manuscript). In summary, we believe that this slight increase in RER could be due to the fact that our experiment was done under resting condition.

3) Based on the hypothesis put forward in this manuscript, a higher expression of HuR would be expected in glycolytic fibers. However, figure 1, panel F suggests increased HuR expression in soleus. How do the authors reconcile these findings?

We thank the reviewer for raising this point. We repeated the western blot shown in Figure 1F and we now show a more representative blot with more appropriate loading, as indicated by Tubulin. Nevertheless, we still see a higher expression levels of HuR in Soleus when compared to other muscle. Since HuR is a ubiquitously expressed protein, we think that under normal conditions, its function (activation or inhibition) is modulated through protein ligand, collaboration, or competition with non-coding RNAs such as micro-RNAs or posttranslational modification. Hence, an increase or decrease in HuR expression levels may not reflect a change in its impact on target mRNAs. This point is now made clear in the discussion section.

4) In the cachexia model, expression of PGC-1 α should be determined to link the proposed mechanism to the reported outcome.

This is an excellent suggestion. We measured PGC-1 α levels in our cachexia experiments and observed that the high levels of PGC-1 α observed in muHuR-KO mice was maintained in the presence LLC tumors when compared to PGC-1 α level in their control counterparts. This experiment is now included as **Figure 8C in the revised manuscript**.

5) The authors claim that increased expression of PPARalpha is linked to the enrichment of type I fibers in skeletal muscle. However, muscle-specific overexpression of PPARalpha results in more glycolytic fibers, whereas muscle-specific deletion of PPARalpha induces an oxidative shift (see Gan et al. 2013 J Clin Invest 123: 2564–2575). The role of PPARalpha in this context thus is much more controversial compared to other important regulators, including PPARdelta, ERRalpha or ERRgamma, that are clearly associated with oxidative muscles. The authors should measure the expression of these and related genes in their paradigm.

We are grateful to the reviewer for catching this oversight on our part. We agree that while the role of PPAR α is not completely defined in muscle fiber type specification, several observations linked its up-regulation with the glycolytic phenotype. Hence, we corrected our statement to reflect this fact. In addition, by performing the experiments suggested by the reviewer we found that PPAR α is indeed down regulated in muHuR-KO muscle (see **Figure 5D and Supplemental Figure 6 of the revised manuscript**). As per the reviewer's prediction, these data agree with Gan *et al.* (Gan *et al.* 2013 J Clin Invest 123: 2564–2575) and are consistent with the oxidative phenotype we report in these animals. In addition, as suggested, we measured the expression levels of the other factors (PPARdelta, ERRalpha or ERRgamma) and did not see any major changes between muHuR-KO muscles and their control counterparts (*data not shown*).

6) Figure 1: Tubulin levels seem higher in HuR KO animals compared to controls?

As mentioned above the western of **Figure 1F** has been redone.

7) Figure 2, panel F: is the difference statistically significant as claimed in the text?

Thanks to the reviewer for this question. The difference is not statistically significant as now shown in Figure 2G of the revised manuscript. The text was corrected accordingly.

8) Why was the number of type IIX not quantified? These would be the unstained fibers.

Thanks to the reviewer for this suggestion and now the number of type IIX is included in **Figure 4**, and **Supplemental Figures 3 and 4**.

9) The authors claim that “the upregulation of PGC-1alpha is associated [...] with the prevention of muscle atrophy induced by several diseases including cancer.” I am not aware of any data showing this. In fact, this seems not to be the case, and PGC-1alpha overexpression has been linked to increased tumor size (see Wang et al. 2012 PLoS ONE 7(3):e33426).

The reviewer is correct in mentioning that the role of PGC-1 α in protecting against cancer-induced muscle atrophy is still debatable. The above-mentioned statement was made based on observations outlined by several papers (Sandri, M. *et al.* *Proc Natl Acad Sci U S A* **103**, 16260-16265 (2006); Cannavino, J. *et al.* *J Physiol* **593**, 1981-1995 (2015); Kang, C. *et al.*, *Ann N Y Acad Sci* **1271**, 110-117 (2012) linking the up-regulation of PGC-1 α levels to protection from atrophy in general. We agree that the connection to cancer-induced atrophy is less evident and still under discussion. Therefore, to reflect this opinion, we amended our statement which no longer makes a reference to cancer.

10) Data from the soleus muscle have been included in the main text (Figures 4 and 5), while the fiber size distribution of other muscles are shown in the supplement, whereas gene expression data of these muscles remain elusive. Even though effects are apparent in the soleus, this muscle might not represent the best model for the experimental context brought forward in this manuscript, since this muscle already is highly oxidative. Thus, even stronger effects of HuR deletion would be expected in mixed and glycolytic muscles, which have a higher window for a shift towards an oxidative phenotype. For example, the absence of changes in the expression of most of the markers determined by qPCR in Figure 5 might be due to the already high basal levels of these in soleus. It thus would be interesting to see analogous data from other muscles.

The reviewer is absolutely correct and our new data somehow support that. We now included data showing gene expression levels in both EDL and Peroneus. These data are now shown as **Supplemental Figures 3D and 4D and 6.**

Reviewers' Comments:

Reviewer #1:

Remarks to the Author:

The authors have satisfactorily addressed all of my substantive concerns.

Reviewer #2:

Remarks to the Author:

This is a revision of previously submitted manuscript that has a strong premise for examining fiber type and the importance HuR regulation in muscle wasting. This has significance as HuR's role in muscle wasting is under explored. While the author is an expert on the regulation of HuR and an impressive number of assays were measured weaknesses were identified.

Overall, the revision has significantly strengthened the manuscript, and the authors have made strong efforts to address this reviewer's prior concerns, which are itemized below.

However, a few minor concerns remain that need to be addressed related to transparency and validity of the results.

1. The authors need to mention in the discussion further directions or limitations the potential for HuR to affect satellite cells and myogenesis, which has been implicated with cachexia.
2. While C2C12 cells are mentioned, the authors need to more clearly state in the results that C2C12 myoblasts and not fully differentiated myotubes. Most of the study infers in vivo that changes are in myofibers and not myogenic precursor cells. However, this needs to be further clarified in the discussion as a future direction or limitation
3. Rigor and reproducibility clarification. In figure 8 the legend states N=4 for A, N=5 for expression levels. However, depending on the treatment group *A shows up to 6 dots for some treatments as does some gene expression. Please clarify why there are different observation numbers and also correct the Ns that are in the legend. This will be noticed.
4. Additionally, related to the new presentation, there are a few important points that need addressed outside the strong phenotype analysis for the KO. the authors need to move Supplemental figure 10B to text figure 8, which is critical in describing the LLC model, and add statistics to the graph. This should be described better in the text. All cachexia researchers know the variability associated with the LLC model and the degree of cachexia in the LLC controls is a requirement for understanding the KO effect.

Related to transparency in the current results, the authors need to state that the LLC control mice in the study never lost body weight (as currently presented). There is a scientific and mechanistic difference between blocked growth and body weight loss from peak bodyweight. Alternatively, the authors could address carcass weight minus tumor weight, which is also commonly done at sacrifice. Do the muscle weight differences in the control and LLC reflect blocked growth or catabolic wasting? Could this be characterized as the initiation of muscle wasting?

Overall, as presented the LLC control mice never lost bodyweight over the study time course. This will not affect the importance of the data but it does frame the effect correctly. The LLC experiment is more about the tumor bearing mouse's cancer environment than blocked growth, as the non-LLC continued to grow during the experimental period but the tumor bearing mice did not grow. This needs to be reflected in the results and discussion.

Itemized responses

- The revision adds additional data in several areas and have successfully addressed this prior concern

Prior Comment: The study impact would be strengthened by more detail in muscle related to the impact of HuR on oxidative metabolism, which is inferred by myosin expression. Additional markers of oxidative enzyme capacity, muscle uncoupling expression would improve the study in the absence of specific muscle function and mitochondrial function.

- The concerns with rigor have been addressed in the revised manuscript, except for figure 8 (see above),

Prior Comment: Several concerns were identified with rigor and reproducibility. The statistical approaches need described. The N for experiments with only 3 or 4 observations needed bolstered . The replication and repeatability strategy of cell culture experiments needs addressed, and details on sampling and quantification or histological and immunohistochemical analysis of muscle needs improved.

-The revision has addressed this concern related to data interpretation

Prior Comment: Page 7 and elsewhere, there are not "non-significant decreases", it is only decreased if the statistical analysis established the change. It is also unclear how controls in Figure 4c have no variation. The controls should still be 1.0 +/- the s.e.m. Figure 5c lacks statistical symbols. Only an N=2 was used for fiber CSA analysis in figure 7 and needs to be increased.

- The authors have thoughtfully addressed this comment, which should be mentioned in the last paragraph of the discussion as a limitation or future direction.

Prior Comment: Do the authors expect the use of MyoD as the muscle specific promoter had any affect on satellite activity? Would this alter any interpretations. Do the authors have data.

- This comment has been successfully addressed

Prior Comment: Abstract, the interpretation of the results is far too strong as the study provides no direct evidence that a fiber type shift protected the KO muscle from wasting. They were just associated. Please edit.

- This comment has been successfully addressed

Prior Comment: Introduction and Abstract: A defined purpose centered on the main variables in the study with and associated hypothesis would improve the impact of the study, as there is currently an open-ended discovery approach which makes the study look much more like a phenotype description after KO. From the mechanisms nicely described in the introduction it looks as if this type of statement could be formulated.

- This comment has been successfully addressed

Prior Comment: Introduction: While comprehensive, paragraphs 1 and 2 have redundancy in concept and should combined.

- This comment has been successfully addressed

Prior Comment: Methods page 20. Please provide the minimum number of muscle fibers quantified per muscle to determine CSA

-The authors need to clearly state in the results that C2I2 myoblasts and not fully differentiated myotubes were analyzed Prior Comment: Methods cell culture. Please specify how long C2C12 myoblasts were allowed to differentiate into myotubes.

-This has been addressed

Prior Comment: Statistical analysis was not included in the methods or figure legends.

Reviewer #3:

Remarks to the Author:

The authors have made a tremendous and laudable effort to address all of my concerns in an adequate manner.

Point-by-point rebuttal to reviewer's 2 comments

We would like to thank all the reviewers for their thoughtful and very helpful comments and suggestions. We are greatfull to all the reviewers for their support of our work.

We tahnk reviewer 2 for the addtionall comments that gave us the opportunity to address all the minor issue identified. This was very helpful in updating the infos related to each experiments outlined in our manuscript.

Below is a copy of reviewer 2 comments and our detailed answers. The reviewer comment are in normal font, and are followed immediatly by our response, highlighted in grey.

Reviewer #2 (Remarks to the Author):

This is a revision of previously submitted manuscript that has a strong premise for examining fiber type and the importance HuR regulation in muscle wasting. This has significance has HuR's role in muscle wasting is under explored. While the author is an expert on the regulation of HuR and an impressive number of assays were measured weaknesses were identified.

Overall, the revision has significantly strengthened the manuscript, and the authors have made strong efforts to address this reviewer's prior concerns, which are itemized below.

We thank the reviewer for the nice comments and the support for our work.

However, a few minor concerns remain that need to be addressed related to transparency and validity of the results.

1. The authors need to mention in the discussion further directions or limitations the potential for HuR to affect satellite cells and myogenesis, which has been implicated with cachexia.

We thank the reviewer for this suggestion that we totally agree with. As suggested we included in paragraph 6 of the discussion section to outline "further directions" on HuR potential role in the commitment of satellite cells to myogenesis. In addition, we also provided some speculation on future direction to assess the impact of HuR on satellite cells function during cachexia.

2. While C2C12 cells are mentioned, the authors need to more clearly state in the results that C2!2 myoblasts and not fully differentiated myotubes. Most of the study infers in vivo that changes are in myofibers and not myogenic precursor cells. However, this needs to be further clarified in the discussion as a future direction or limitation

We are grateful and thankfult to the reviewer for catching this oversight on our part. We revised our main text (result and discussion sections), methods and figure legedends to make it clear that we used C2C12 myoblasts in the experiments outlined in Figures 6 and 7 as well as in Supplementary Figures 8 and 9.

3. Rigor and reproducibility clarification. In figure 8 the legend states N=4 for A, N=5 for expression levels. However, depending on the treatment group *A shows up to 6 dots for some treatments as does some gene expression. Please clarify why there are different observation numbers and also correct the Ns that are in the legend. This will be noticed.

We thank the reviewer for raising this issue. We amended the legends of each panel in Figure 8 and reported the exact number of animals used for each group. In fact we also updated these same information in all our figure legends in the main and supplementary figures. In addition, we provide the Source data for the majority of the experiments reported in both the main and Supplementary Figures.

4. Additionally, related to the new presentation, there are a few important points that need addressed outside the strong phenotype analysis for the KO. the authors need to move Supplemental figure 10B to text figure 8, which is critical in describing the LLC model, and add statistics to the graph. This should be described better in the text. All cachexia researchers know the variability associated with the LLC model and the degree of cachexia in the LLC controls is a requirement for understanding the KO effect.

Related to transparency in the current results, the authors need to state that the LLC control mice in the study never lost body weight (as currently presented). There is a scientific and mechanistic difference between blocked growth and body weight loss from peak bodyweight. Alternatively, the authors could address carcass weight minus tumor weight, which is also commonly done at sacrifice. Do the muscle weight differences in the control and LLC reflect blocked growth or catabolic wasting? Could this be characterized as the initiation of muscle wasting?

Overall, as presented the LLC control mice never lost bodyweight over the study time course. This will not affect the importance of the data but it does frame the effect correctly. The LLC experiment is more about the tumor bearing mouse's cancer environment that blocked growth, as the non-LLC continued to grow during the experimental period but the tumor bearing mice did not grow. This needs to be reflected in the results and discussion.

The reviewer is correct and we agree with his suggestions. In fact to address his concern we did the following:

- **In figure 8**, we included a new panel 8a to report "carcass weight minus tumor weight at sacrifice", the measurement we have already collected from the animals reported in the previous Supplementary Figure 10B (that is now shown as Supplementary Figure 10a). The results clearly show that when the tumor is removed at sacrifice the LLC-muHuR-KO animals were protected from LLC-induced muscle loss when compared to their control counterparts.
- We moved Supplemental Figure 10b to become Supplemental Figure 10a.
- The figure legends of both Figure 8 and Supplemental Figure 10 as well as the result section of the main text were amended to reflect the above mentioned changes.

Itemized responses

- The revision adds additional data in several areas and have successfully addressed this prior concern
Prior Comment: The study impact would be strengthened by more detail in muscle related to the

impact of HuR on oxidative metabolism, which is inferred by myosin expression. Additional markers of oxidative enzyme capacity, muscle uncoupling expression would improve the study in the absence of specific muscle function and mitochondrial function.

- The concerns with rigor have been addressed in the revised manuscript, except for figure 8 (see above),

Prior Comment: Several concerns were identified with rigor and reproducibility. The statistical approaches need described. The N for experiments with only 3 or 4 observations needed bolstered . The replication and repeatability strategy of cell culture experiments needs addressed, and details on sampling and quantification or histological and immunohistochemical analysis of muscle needs improved.

-The revision has addressed this concern related to data interpretation

Prior Comment: Page 7 and elsewhere, there are not “non-significant decreases”, it is only decreased if the statistical analysis established the change. It is also unclear how controls in Figure 4c have no variation. The controls should still be 1.0 +/- the s.e.m. Figure 5c lacks statistical symbols. Only an N=2 was used for fiber CSA analysis in figure 7 and needs to be increased.

- The authors have thoughtfully addressed this comment, which should be mentioned in the last paragraph of the discussion as a limitation or future direction.

Prior Comment: Do the authors expect the use of MyoD as the muscle specific promoter had any affect on satellite activity? Would this alter any interpretations. Do the authors have data.

As mentioned above a statement in paragraph 6 of the discussion has been added to address this issue.

- This comment has been successfully addressed

Prior Comment: Abstract, the interpretation of the results is far too strong as the study provides no direct evidence that a fiber type shift protected the KO muscle from wasting. They were just associated. Please edit.

- This comment has been successfully addressed

Prior Comment: Introduction and Abstract: A defined purpose centered on the main variables in the study with and associated hypothesis would improve the impact of the study, as there is currently an open-ended discovery approach which makes the study look much more like a phenotype description after KO. From the mechanisms nicely described in the introduction it looks as if this type of statement could be formulated.

- This comment has been successfully addressed

Prior Comment: Introduction: While comprehensive, paragraphs 1 and 2 have redundancy in concept and should combined.

- This comment has been successfully addressed

Prior Comment: Methods page 20. Please provide the minimum number of muscle fibers quantified

per muscle to determine CSA

- The authors need to clearly state in the results that C212 myoblasts and not fully differentiated myotubes were analyzed
Prior Comment: Methods cell culture. Please specify how long C2C12 myoblasts were allowed to differentiate into myotubes.

As mentioned above this issue has been addressed throughout the manuscript and figures legends.

-This has been addressed

Prior Comment: Statistical analysis was not included in the methods or figure legends.